# Relaxing restrictions at the pace of vaccination increases freedom and guards against further COVID-19 waves

**Simon Bauer**[1,☯], **Sebastian Contreras**[1,2,☯], **Jonas Dehning**[1], **Matthias Linden**[1,3], **Emil Iftekhar**[1], **Sebastian B. Mohr**[1], **Alvaro Olivera-Nappa**[2], **Viola Priesemann**[1,4]*

**1** Max Planck Institute for Dynamics and Self-Organization, Göttingen, Germany, **2** Centre for Biotechnology and Bioengineering, Universidad de Chile, Santiago, Chile, **3** Institute for Theoretical Physics, Leibniz University, Hannover, Germany, **4** Institute for the Dynamics of Complex Systems, University of Göttingen, Göttingen, Germany

☯ These authors contributed equally to this work.
* viola.priesemann@ds.mpg.de

**Data Availability Statement:** The source code for data generation and analysis is available online on GitHub https://github.com/Priesemann-Group/covid19_vaccination. All other relevant data are

## Abstract

Mass vaccination offers a promising exit strategy for the COVID-19 pandemic. However, as vaccination progresses, demands to lift restrictions increase, despite most of the population remaining susceptible. Using our age-stratified SEIRD-ICU compartmental model and curated epidemiological and vaccination data, we quantified the rate (relative to vaccination progress) at which countries can lift non-pharmaceutical interventions without overwhelming their healthcare systems. We analyzed scenarios ranging from immediately lifting restrictions (accepting high mortality and morbidity) to reducing case numbers to a level where test-trace-and-isolate (TTI) programs efficiently compensate for local spreading events. In general, the age-dependent vaccination roll-out implies a transient decrease of more than ten years in the average age of ICU patients and deceased. The pace of vaccination determines the speed of lifting restrictions; Taking the European Union (EU) as an example case, all considered scenarios allow for steadily increasing contacts starting in May 2021 and relaxing most restrictions by autumn 2021. Throughout summer 2021, only mild contact restrictions will remain necessary. However, only high vaccine uptake can prevent further severe waves. Across EU countries, seroprevalence impacts the long-term success of vaccination campaigns more strongly than age demographics. In addition, we highlight the need for preventive measures to reduce contagion in school settings throughout the year 2021, where children might be drivers of contagion because of them remaining susceptible. Strategies that maintain low case numbers, instead of high ones, reduce infections and deaths by factors of eleven and five, respectively. In general, policies with low case numbers significantly benefit from vaccination, as the overall reduction in susceptibility will further diminish viral spread. Keeping case numbers low is the safest long-term strategy because it considerably reduces mortality and morbidity and offers better preparedness against emerging escape or more contagious virus variants while still allowing for higher contact numbers (freedom) with progressing vaccinations.

within the manuscript and its Supporting information files.

**Funding:** SB, SC, JD, ML, EI, SM, and VP received support from the Max-Planck-Gesellschaft (MPRG Priesemann), https://www.mpg.de/de. SC and AO-N received support from the Comisión Nacional de Investigación Científica y Tecnológica PIA project FB0001, ANID, Chile. ML, JD, SM acknowledge funding from the "Netzwerk Universitätsmedizin" (NUM) project egePan (01KX2021). The funders had no role in study design, data collection and analysis, decision to publish, or preparation of the manuscript.

**Competing interests:** The authors have declared that no competing interests exist.

## Author summary

In this work, we quantify the rate at which non-pharmaceutical interventions can be lifted as COVID-19 vaccination campaigns progress. With the constraint of not exceeding ICU capacity, there exists only a relatively narrow range of plausible scenarios. We selected different scenarios ranging from the immediate release of restrictions to more conservative approaches aiming at low case numbers. In all considered scenarios, the increasing overall immunity (due to vaccination or post-infection) will allow for a steady increase in contacts. However, deaths and total cases (potentially leading to *long covid*) are only minimized when aiming for low case numbers, and restrictions are lifted at the pace of vaccination. These qualitative results are general. Taking EU countries as quantitative examples, we observe larger differences only in the long-term perspectives, mainly due to varying seroprevalence and vaccine uptake. Thus, the recommendation is to keep case numbers as low as possible to facilitate test-trace-and-isolate programs, reduce mortality and morbidity, and offer better preparedness against emerging variants, potentially escaping immune responses. Keeping moderate preventive measures in place (such as improved hygiene, use of face masks, and moderate contact reduction) is highly recommended will further facilitate control.

## Introduction

The rising availability of effective vaccines against SARS-CoV-2 promises the lifting of restrictions, thereby relieving the social and economic burden caused by the COVID-19 pandemic. However, it is unclear how fast the restrictions can be lifted without risking another wave of infections; we need a promising long-term vaccination strategy [1]. Nevertheless, a successful approach has to take into account several challenges; vaccination logistics and vaccine allocation requires a couple of months [2–4], vaccine eligibility depends on age and eventually serostatus [5], vaccine acceptance may vary across populations [6], and more contagious [7] and escape variants of SARS-CoV-2 that can evade existing immunity [8, 9] may emerge, thus posing a persistent risk. Last but not least, disease mitigation is determined by how well vaccines block infection, and thus prevent the propagation of SARS-CoV-2 [3, 4], the time to develop effective antibody titers after vaccination, and their efficacy against severe symptoms. All these parameters will greatly determine the design of an optimal strategy for the transition from epidemicity to endemicity [10].

To bridge the time until a significant fraction of the population is vaccinated, a sustainable public health strategy has to combine vaccination with non-pharmaceutical interventions (NPIs). Otherwise it risks further waves and, consequently, high morbidity and excess mortality. However, the overall compliance with NPIs worldwide has on average decreased due to a "pandemic-policy fatigue" [11]. Therefore, the second wave has been more challenging to tame [12] although NPIs, in principle, can be highly effective, as seen in the first wave [13, 14]. After vaccinating the most vulnerable age groups, the urge and social pressure to lift restrictions will increase. However, given the wide distribution of fatalities over age groups and the putative incomplete protection of vaccines against severe symptoms and against transmission, NPIs cannot be lifted entirely or immediately. With our study, we want to outline at which pace restrictions can be lifted as the vaccine roll-out progresses.

Public-health policies in a pandemic have to find a delicate ethical balance between reducing the viral spread and restricting individual freedom and economic activities. However, the interest of health on the one hand and society and economy on the other hand are not always

contradictory. For the COVID-19 pandemic, all these aspects clearly profit from low case numbers [15–17], i.e., an incidence where test-trace-and-isolate (TTI) programs can efficiently compensate for local spreading events. The challenge is to reach low case numbers and maintain them [18, 19]. Especially with the progress of vaccination, restrictions should be lifted when the threat to public health is reduced. However, the apparent trade-off between public health interest and freedom is not always linear and straightforward. Taking into account that low case numbers facilitate TTI strategies (i.e., health authorities can concentrate on remaining infection chains and stop them quickly) [18–20], an optimal strategy with a low public health burden *and* large freedom may exist and be complementary to vaccination.

Here, we quantitatively study how the planned vaccine roll-out in the European Union (EU), together with the cumulative post-infection immunity (seroprevalence), progressively allows for lifting restrictions. In particular, we study how precisely the number of contacts can be increased without rendering disease spread uncontrolled over the year 2021. Our study builds on carefully curated epidemiological and contact network data from Germany, France, the UK, and other European countries. Thereby, our work can serve as a blueprint for an opening strategy.

## Analytical framework

Our analytical framework builds on our deterministic, age-stratified, SEIRD-ICU compartmental model, modified to incorporate vaccination through delay differential equations. It includes compartments for a 2-dose staged vaccine roll-out, immunization delays, intensive care unit (ICU)-hospitalized, and deceased individuals. A central parameter for our model is the gross reproduction number $R_t$. It is essentially the time-varying effective reproduction number without considering the effects of immunity nor of TTI. That number depends (among several factors) on i) the absolute number of contacts per individual, and ii) the probability of being infected given a contact. In other words, $R_t$ is defined as the average number of contacts an infected individual has that would lead to an offspring infection in a fully susceptible population. Therefore, an increase in $R_t$ implies an increase in contact frequency or the probability of transmission per contact, e.g., due to less mask-wearing. The core idea is that increasing immunity levels among the population (post-infection or due to vaccination) allows for a higher average number of potentially contagious contacts and, thus, freedom (quantified by $R_t$), given the same level of new infections or ICU occupancy. Hence, with immunization progress reducing the susceptible fraction of the population, $R_t$ can be dynamically increased while maintaining control over the pandemic, i.e., while keeping the effective reproduction number below one (Fig 1A).

To adapt the gross reproduction number $R_t$ such that a specific strategy is followed (e.g. staying below TTI or ICU capacity), we include an automatic, proportional-derivative (PD) control system [21]. This control system allows for steady growth in $R_t$ as long as it does not lead to overflowing ICUs (or surpassing the TTI capacity). However, when risking surpassing the ICU capacity, restrictions might be tightened again. In that way, we approximate the feedback-loop between political decisions, people's behavior, reported case numbers, and ICU occupancy.

The basic reproduction number is set to $R_0 = 4.5$, reflecting the dominance of the B.1.1.7 variant [3, 7]. We further assume that the reproduction number can be decreased to about 3.5 by hygiene measures, face masks, and mild social distancing. This number is informed by the estimates of Sharma et al. [22], who estimate the combined effectiveness of mask wearing, limiting gatherings to at most 10 people and closing night clubs to a reduction of about 20–40%, thus leading to a reproduction number between 2.7 and 3.6. We use a conservative estimate, as

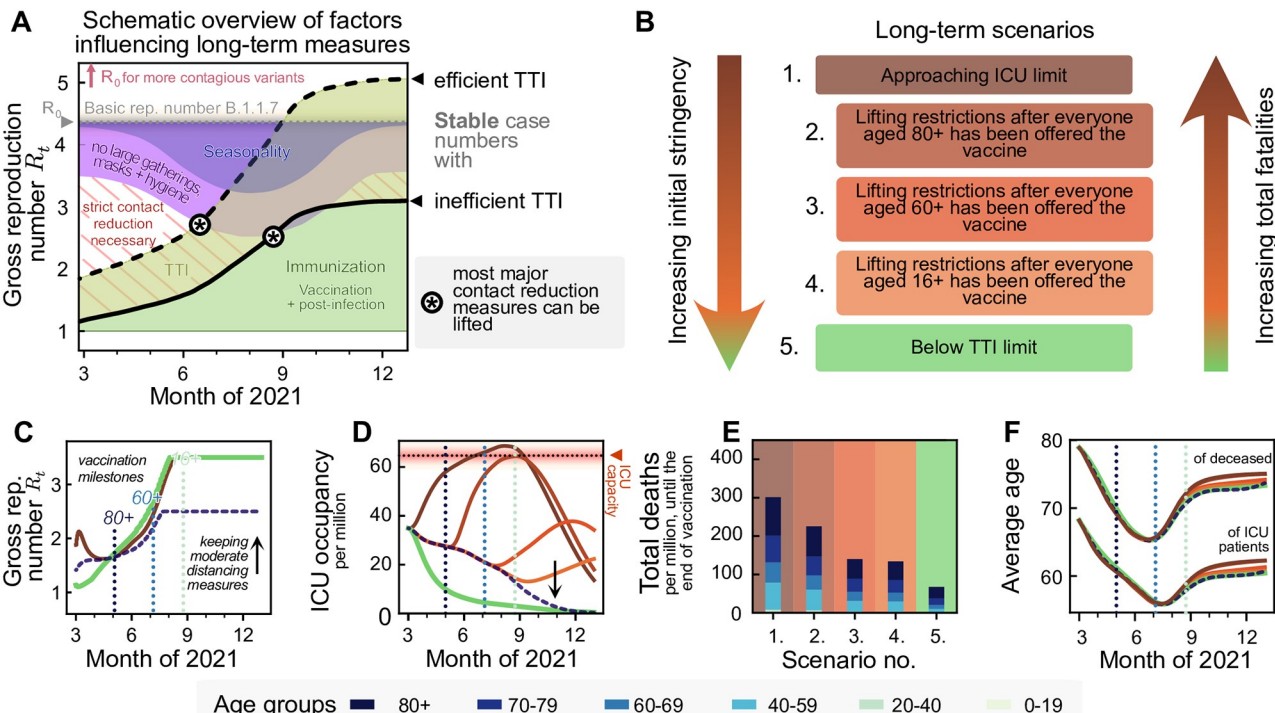

**Fig 1. With progressing vaccination in the European Union, a slow but steady increase in freedom will be possible. However, premature lifting of NPIs considerably increases the total fatalities without a major reduction in restrictions in the middle term. A**: A schematic outlook into the effect of vaccination on societal freedom. Freedom is quantified by the maximum time-varying gross reproduction number ($R_t$) allowed to sustain stable case numbers. As $R_t$ does not consider the immunized population, gross reproduction numbers above one are possible without rendering the system unstable. A complete return to pre-pandemic behavior would be achieved when $R_t$ reaches the value of the basic reproduction number $R_0$ (or possibly at a lower value due to seasonality effects during summer, purple-blue shaded area). The thick full and dashed lines indicate the gross reproduction number $R_t$ allowed to sustain stable case numbers if test-trace-isolate (TTI) programs are inefficient and efficient, respectively, which depends on the case numbers level. Increased population immunity (green) is expected to allow for lifting the most strict contact reduction measures while only keeping mild NPIs (purple) during summer 2021 in the northern hemisphere. Note that seasonality is not explicitly modeled in this work. See S4 Fig for an extended version including the year 2020. **B**: We explore five different scenarios for lifting restrictions in the EU, in light of the EU-wide vaccination programs. We sort them according to the initial stringency that they require and the total fatalities that they may cause. One extreme (Scenario 1) offers immediate (but still comparably little) freedom by approaching ICU-capacity limits quickly. The other extreme (Scenario 5) uses a strong initial reduction in contacts to allow long-term control at low case numbers. Finally, the intermediate scenarios initially maintain moderate case numbers and lift restrictions at different points in the vaccination program. **C**: All extreme strategies allow for a steady noticeable increase in contacts in the coming months (cf. panel **A**), but vary greatly in the (**D**) ICU-occupancy profiles and (**E**) total fatalities. **F**: Independent on the strategy, we expect a transient but pronounced decrease in the average age of ICU patients and deceased over the summer.

this is only a exemplary set of restrictions. Therefore, we restrict $R_t$ in general not to exceed 3.5 (Fig 1C).

Efficient TTI contributes to reducing the effective reproduction number. Hence, it increases the average number of contacts (i.e., $R_t$) that people may have under the condition that case numbers remain stable (Fig 1A) [18]. This effect is particularly strong at low case numbers, where the health authorities can concentrate on tracing every case efficiently [19]. Here, we approximate the effect of TTI on $R_t$ semi-analytically to achieve an efficient implementation (see Methods).

For vaccination, we use as default parameters an average vaccine efficacy of 90% protection against severe illness [23] and of 75% protection against infection [24]. We further assume that vaccinated individuals with a breakthrough infection carry a lower viral load and thus are 50% less infectious [25] than unvaccinated infected individuals. We assume a total average vaccine uptake of 80% [26] that increases with age from 73% in the 0–19 to 89% in the 80+ age group, and an age-prioritized vaccine delivery as described in the Methods section. In detail, most of

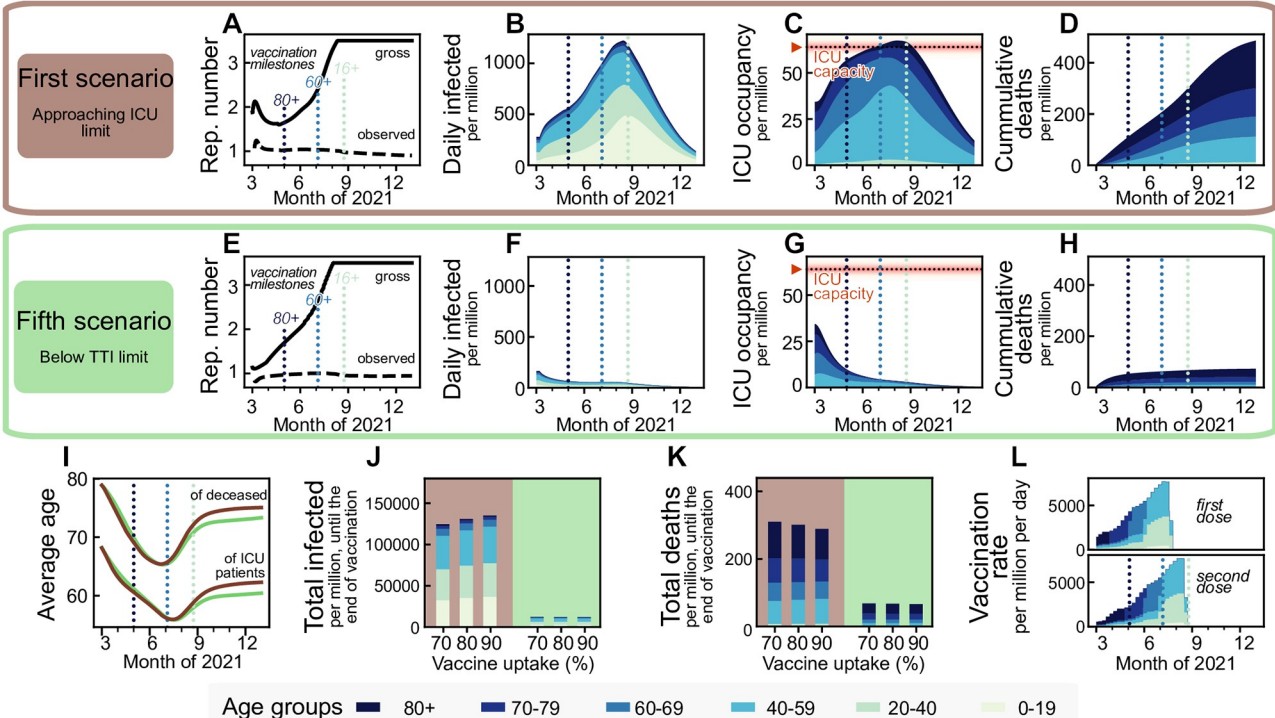

**Fig 2. Maintaining low case numbers during vaccine roll-out reduces the number of ICU patients and deaths by about a factor five compared to quickly approaching the ICU limit while hardly requiring stronger restrictions.** Aiming to maximize ICU occupancy (**A–D**) allows for a slight increase of the allowed gross reproduction number $R_t$ early on, whereas lowering case numbers below the TTI capacity limit (**E–H**) requires comparatively stronger initial restrictions. Afterwards, the vaccination progress allows for a similar increase in freedom (quantified by increments in $R_t$) for both strategies, starting approximately in May 2021. **B–D, F–H**: These two strategies lead to a completely different evolution of case numbers, ICU occupancy, and cumulative deaths, but differ only marginally in the evolution of the average age of deceased and ICU patients (**I**), as the latter is rather an effect of the age-prioritized vaccination than of a particular strategy. **J,K**: The total number of cases until the end of the vaccination period (of the 80% uptake scenario, i.e., end of August, the rightmost dotted light blue line in sub-panels **A–H**) differ by a factor of eleven between the two strategies, and the total deaths by a factor of five. Vaccine uptake (i.e., the fraction of the eligible, 16+, population that gets vaccinated) has a minor impact on these numbers until the end of the vaccine roll-out but determines whether a wave would follow afterward (see below). **L**: Assumed vaccination rate as projected for Germany, which is expected to be similar across the European Union. For a full display of the time-evolution of the compartments for different uptakes see S6–S8 Figs.

the vaccines are distributed first to the age group 80+, then 70+, 60+, and then to anyone of age 16+. A small fraction of the weekly available vaccines is distributed randomly (e.g. because of profession). After everyone got a vaccine offer roughly by the end of August, we assume no further vaccination (see Fig 2L). The daily amount of vaccine doses per million is derived from German government projections, but is expected to be similar across the EU. For the course of the disease, the age-dependent fraction of non-vaccinated, infected individuals requiring intensive care is estimated from German hospitalization data, using the infection-fatality-ratio (IFR) reported in [27] (see Table 1 and Methods).

In our default scenario we use a contact structure between age groups as measured during pre-pandemic times [28]. However, we halve the infection probability in the 0–19 year age group to account for reduced in-person classes and better ventilation and systematic random screening in school settings using rapid COVID-19 tests. Under these assumptions, the infection probability among the 0–19 age group is similar to the one among the 20–39 and 40–69 age groups. We start our simulations at the beginning of March 2021, with an incidence of 200 daily infections per million, two daily deaths per million, an ICU occupancy of 30 patients per million, a seroprevalence of 10%, and about 4% of the population already vaccinated. This is

**Table 1. Age-dependent infection-fatality-ratio (IFR), probability of requiring intensive care due to the infection (ICU probability) and ICU fatality ratio (ICU-FR).** The IFR is defined as the probability of an infected individual dying, whereas the ICU-FR is defined as the probability of an infected individual dying while receiving intensive care.

| Age | IFR [27] | ICU probability | ICU-FR | Avg. ICU time (days) |
|---|---|---|---|---|
| 0–19 | 0.00002 | 0.00014 | 0.0278 | 5 |
| 20–39 | 0.00022 | 0.00203 | 0.0389 | 5 |
| 40–59 | 0.00194 | 0.01217 | 0.0678 | 11 |
| 60–69 | 0.00739 | 0.04031 | 0.1046 | 11 |
| 70–79 | 0.02388 | 0.05435 | 0.1778 | 9 |
| >80 | 0.08292 | 0.07163 | 0.4946 | 6 |
| Average | 0.00957 | 0.02067 | 0.0969 | 9 |

comparable to German data (assuming a case under-reporting factor of 2, which had been measured during the first wave in Germany [29]) and typical for EU countries at the beginning of March 2021 (further details in the Methods section). We furthermore explore the impact of important differences between EU countries, namely the seroprevalance by the start of the vaccination program, demographics, and vaccine uptake exemplary for Finland, Italy and the Czech Republic in addition to the default German parameters.

## Results

### Aiming for low case numbers has the best long-term outcome

We first present the two extreme scenarios: case numbers quickly rise so that the ICU capacity limit is approached (Scenario 1), or case numbers quickly decline below the TTI capacity limit (Scenario 5; Fig 2). We set the ICU capacity limit at 65 patients/million, reflecting the maximal occupancy and improved treatments during the second wave in Germany [30] and use German demographics. The incidence (daily new cases) limit up to which TTI is fully efficient is set to 20 daily infections per million [15], but depends strongly on the gross reproduction number, as described in Methods.

The first scenario ('approaching ICU limit', Fig 2A–2D) maximizes the initial freedom individuals might have (quantified as the allowed gross reproduction number $R_t$). However, the gained freedom is only transient as, once ICUs approach their capacity limit, restrictions need to be tightened (Fig 2J and 2K). Additionally, stabilization at high case numbers leads to many preventable fatalities, especially in light of likely temporary overflows of the ICU capacity due to the hard-to-control nature of high case numbers.

The fifth scenario ('below TTI limit', Fig 2E and 2F) requires maintaining stronger restrictions for about two months to lower case numbers below the TTI capacity. Afterward, the progress of the vaccination allows for a steady increase in $R_t$ while keeping case numbers low, enabling TTI to contribute to the containment effectively. From May 2021 on, this fifth scenario would allow for slightly more freedom, i.e., a higher $R_t$, than the first scenario (Fig 1C). Furthermore, this scenario reduces morbidity and mortality: Deaths until the of the vaccination period (end of August) are reduced by a factor of five, total infections even by a factor of eleven. Due to the prioritization of the elderly in vaccination, the average age of ICU patients and fatalities drops by roughly 12 and 15 years, respectively, independently of the choice of scenarios (Fig 2I). Overall, the low-case-number scenario thus allows for a very similar increase in freedom over the whole time frame (quantified as the increase in $R_t$) and implies about fives times fewer deaths by the end of the vaccination program compared to the first scenario with high case numbers (Fig 2K).

The vaccine uptake has little influence on the number of deaths and total cases during the vaccination period (Fig 2J and 2K), mainly because restrictions are quickly enacted when reaching the ICU capacity. However, uptake becomes a crucial parameter; It controls the pandemic progression after completing the vaccine roll-out as it determines the residual susceptibility of the population (cf. below). With insufficient vaccination uptake, a novel wave will follow as soon as restrictions are lifted [3].

## Maintaining low case numbers at least until vulnerable groups (60+) are vaccinated is necessary to prevent a severe further wave

Between the two extreme scenarios 1 and 5, which respectively allow maximal or minimal initial freedom, we explore three alternative scenarios, where the vaccination progress and the slow restriction lifting roughly balance out (Figs 3 and 1B). These scenarios assume approximately constant case numbers and then a swift lifting of most of the remaining restrictions within a month after three different vaccination milestones: when the age group 80+ has been vaccinated (Scenario 2, Fig 3A–3D), when the age groups 60+ has been vaccinated (Scenario 3, Fig 3E–3H) and when the entire adult population (16+) has been vaccinated (Scenario 4, Fig 3I–3L).

The relative freedom gained by lifting restrictions early in the vaccination timeline (Scenario 2) hardly differs from the freedom gained from the other two scenarios (Fig 3M), as since new contact restrictions need to take place once reaching the ICU capacity limit, and the initial freedom is partly lost. Significantly, lifting restrictions later reduces the number of infections and deaths by more than 50% and 35% respectively if case numbers have been kept at a moderate level (250 daily infections per million) and by more than 85% and 65%, respectively if case numbers have been kept at a low level (50 per million) beforehand (Fig 3N and 3O). Lifting restrictions entirely after either offering vaccination to everyone aged 60+ or everyone aged 16+ only changes the total fatalities by a small amount, mainly because the vaccination pace is planned to be quite fast by then, and the 60+ age brackets make up the bulk of the highest-risk groups. Hence, a potential subsequent wave only unfolds after the end of the planned vaccination campaign (Fig 3F and 3H). Thus, with the current vaccination plan, it is recommended to keep case numbers at moderate or low levels, at least until the population at risk and people of age 60+ have been vaccinated.

If maintaining low or intermediate case numbers in the initial phase, vaccination starts to decrease the ICU occupancy considerably in May 2021 (Fig 3G and 3K). However, This decrease in ICU occupancy must not be mistaken for a generally stable situation. As soon as restrictions are relaxed too quickly, ICU occupancy surges again (Fig 3C, 3G and 3H), without any relevant gain in freedom for the total population. Nonetheless, the progress in vaccination will, in any case, allow lifting restrictions gradually.

## The long-term success of the vaccination campaign strongly depends on vaccine uptake and vaccine efficacy

The vaccination campaign's long-term success will depend on both people's vaccine uptake and the efficacy of the vaccine against those variants of SARS-CoV-2 prevalent at the time of writing of this paper. A vaccine's efficacy has two contributions: first, vaccinated individuals become less likely to develop severe symptoms and require intensive care [31–33] (vaccine efficacy, $\kappa$). Second, a fraction $\eta$ of vaccinated individuals gains sterilizing immunity, i.e., is completely protected against infections and does not contribute to viral spread at all [24, 34]. We also assume that breakthrough infections among vaccinated individuals would bear lower viral loads, thus exhibit reduced transmissibility [25] (reduced viral load, $\sigma$). However, the

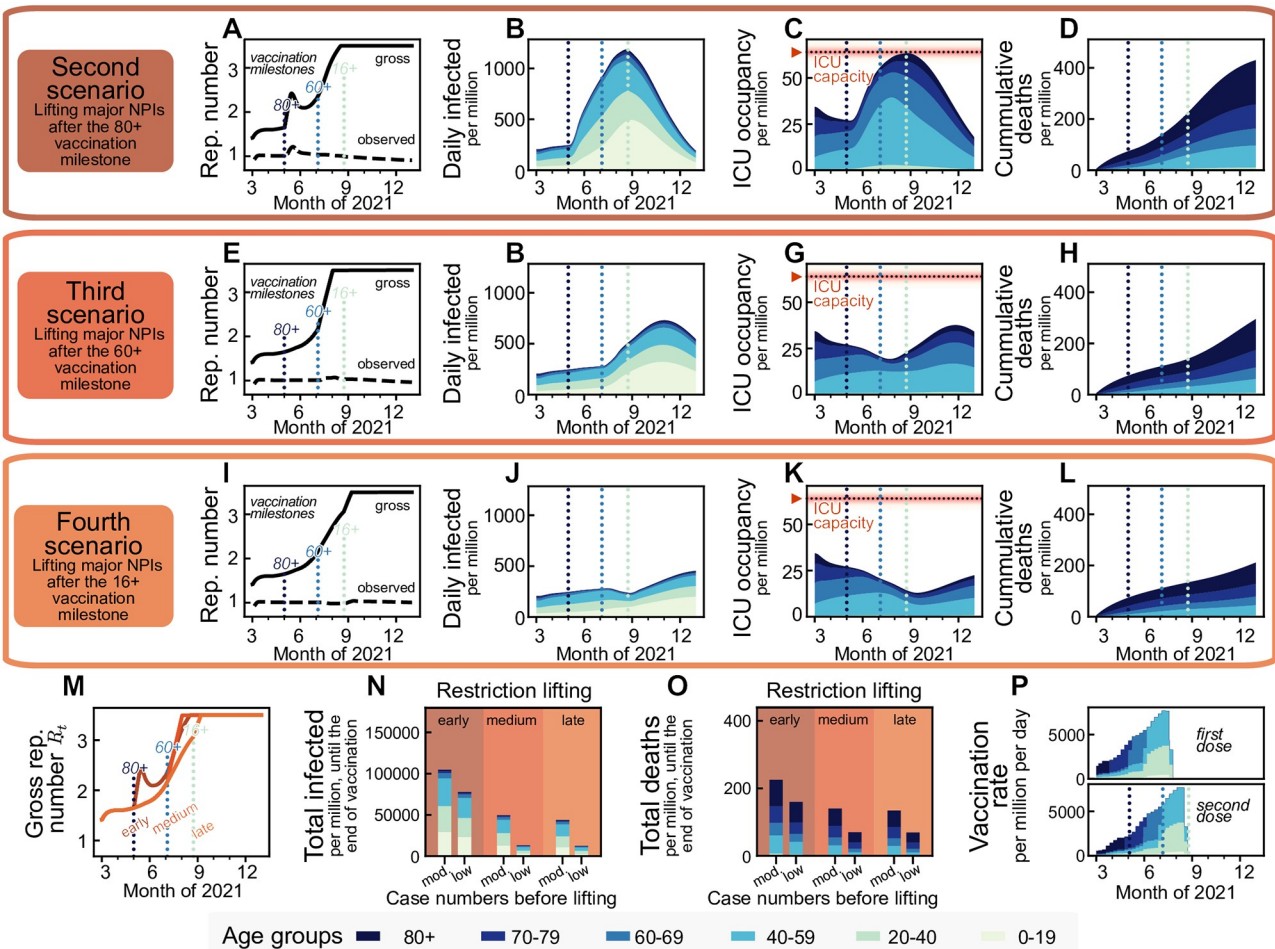

**Fig 3. Vaccination offers a steady return to normality until the end of summer 2021 in the northern hemisphere, no matter whether a transient easing of restrictions is allowed earlier or later (second and fourth scenario, respectively). However, lifting restrictions later reduces fatalities by more than 35%.** We assume that the vaccine immunization progress is balanced out by a slow lifting of restrictions, keeping case numbers at a moderate level ($\leq$ 250 daily new cases per million people). We simulated lifting all restrictions within a month starting from different time points: when (**A–D**) the 80+ age group, (**E–H**) the 60+ age group or (**I–L**) everyone 16+ has been offered vaccination. Restriction lifting leads to a new surge of cases in all scenarios. New restrictions are put in place if ICUs would otherwise collapse. **M**: Lifting all restrictions too early increases the individual freedom only temporarily before new restrictions have to be put in place to avoid overwhelming ICUs. Overall, trying to lift restrictions earlier has a small influence on the additional increase in the allowed gross reproduction number $R_t$. **N,O**: Relaxing major restrictions only medium-late or late reduces fatalities by more than 35% and infections by more than 50%. Fatalities and infections can be cut by an additional factor of more than two when aiming for a *low* (50 per million) instead of *moderate* (250 per million) level of daily infections before major relaxations. **P**: Assumed daily vaccination rates, same as in Fig 2.

possibly reduced effectiveness of vaccines against current variants of concern (VOCs), e.g., B.1.351 and P.1 [32, 35, 36], and potential future VOCs render long-term scenarios about the success of vaccination uncertain.

Therefore, we explore different parameters of vaccine uptake and effectiveness. We quantify the success, or rather the lack of success of the vaccination campaign by the duration of the period where ICUs function near capacity limit, until population immunity is reached. Two different scenarios are considered upon finishing the vaccination campaign: in the first scenario, most restrictions are lifted, like in the previous scenarios (Fig 4B). In the second, restrictions are only lifted partially, to a one third lower gross reproduction number ($R_t$ = 2.5) (Fig 4C). This second scenario presents the long-term maintenance of moderate social distancing

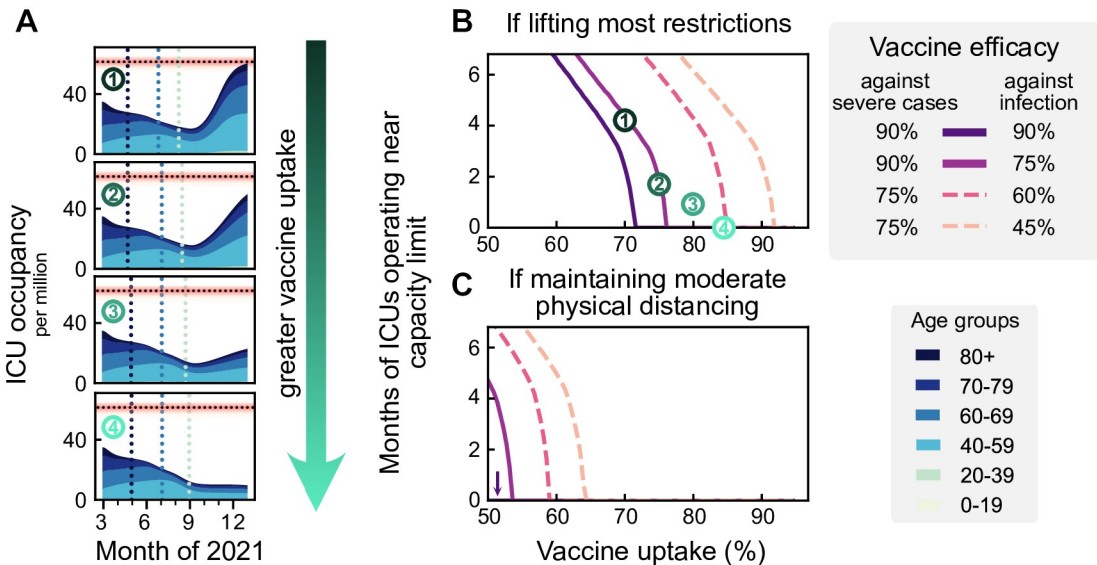

**Fig 4. A high vaccine uptake (> 90% or higher among the eligible population) is crucial to prevent a wave when lifting restrictions after completing vaccination campaigns. A**: We assume that infections are kept stable at 250 daily infections until all age groups have been vaccinated. Then restrictions are lifted, leading to a wave if the vaccine uptake has not been high enough (top three plots). **B**: The duration of the wave (measured by the total time that ICUs function close to their capacity limit) depends on vaccine uptake and vaccine efficacy. We explore the dependency on the efficacy both for preventing severe cases (full versus dashed lines) and preventing infection (shades of purple). The dashed lines might correspond to vaccine efficacy in the event of the emergence of escape variants of SARS-CoV-2. **C**: If some NPIs remain in place (such that the gross reproduction number stays at $R_t = 2.5$), ICUs will not overflow even if the protection against infection is only around 60%. See S2 Fig for all possible combinations of vaccine efficacies, also in the event of different contact structures.

measures, including the restriction of large gatherings to smaller than 100 people, encouraging home-office, enabling effective test-trace-and-isolate (TTI) programs at very low case numbers, and supporting hygiene measures and face mask usage. Fig 4B and 4C indicates how long ICUs are expected to be full in both scenarios, and for different parameters of vaccine efficacy (which may account for the emergence of vaccine escape variants).

The primary determinant for the success of vaccination programs after lifting most restrictions is the vaccine uptake among the population aged 20+; only with a high vaccine uptake (> 90%) we can avoid a novel wave of full ICUs (default parameters as in scenario 3; Fig 4B, $\kappa = 90\%$, $\eta = 75\%$). However, if vaccine uptake was lower or vaccines prove to be less effective against prevalent or new variants, lifting most restrictions would imply that ICUs will work at the capacity limit for months.

In contrast, maintaining moderate social distancing measures (Fig 4C) may prevent a wave after completing the vaccine roll-out. This strategy can also compensate for a low vaccine uptake, requiring only about 55% uptake to avoid surpassing ICU capacity for our default parameters. Nonetheless, any increase in vaccine uptake lowers intensive care numbers, increases freedom, and most importantly, provides better protection in case of the emergence of escape variants, as this would involve an effective reduction of vaccine efficacy (dashed lines). A full exploration of vaccine efficacy parameter combinations and different contact structures is presented in S2 Fig.

Heterogeneity among countries on an EU-wide level will affect the probability and strength of a new wave after completing vaccination campaigns. We chose some exemplary European countries to investigate how our results depend on age demographics, contact structure, and the degree of initial post-infection immunization (seroprevalence). We obtained the

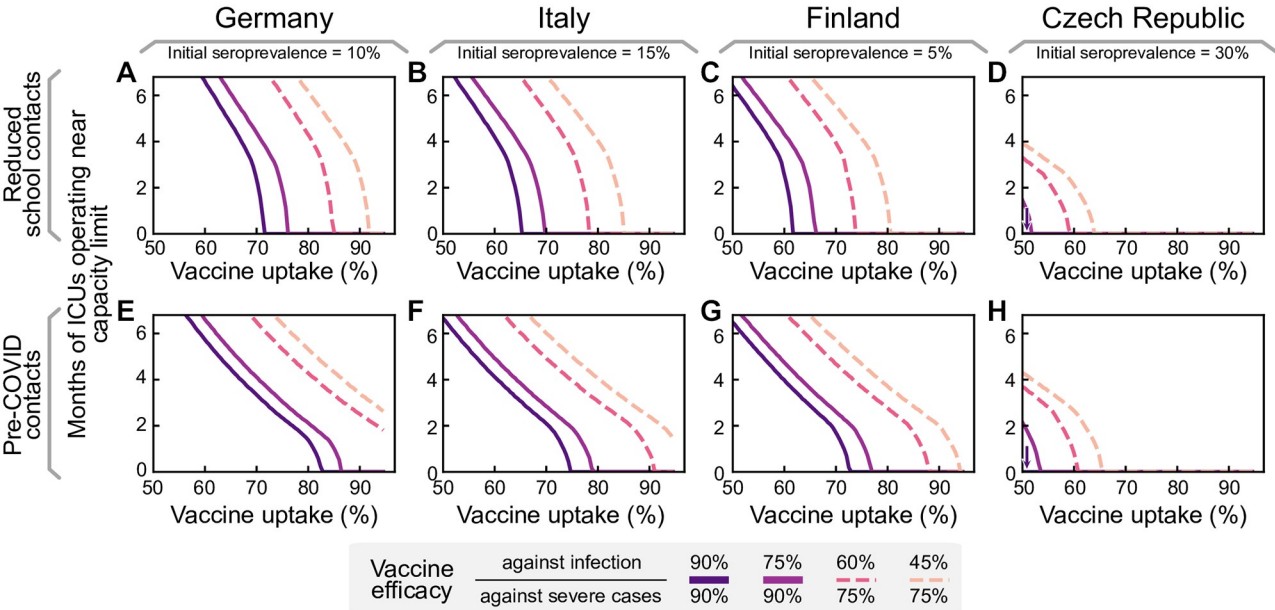

**Fig 5. Seroprevalence and different demographics across EU countries determine the vaccine uptake required for population immunity.** As in Fig 4B, we assume that case numbers are stable at 250 daily infections per million per day until the end of vaccination, when most restrictions are lifted (such that the gross reproduction number goes up to 3.5). We vary the initial seroprevalence and age demographics and contact structures to represent German, Italian, Finnish, and Czech data. **A–D**: Projected ICU occupancy in a subsequent wave depending on vaccine uptake, assuming reduced transmission risk in schools but otherwise default pre-pandemic contact structures. **E–H**: Projected ICU occupancy depending on vaccine uptake, assuming default pre-pandemic contact structures everywhere (including schools). See S3 Fig for a more comprehensive exploration of combinations of vaccine efficacies.

seroprevalence in the different countries by scaling the German 10% seroprevalence with the relative differences in cumulative reported case numbers between Germany and the other countries, i.e., we assume the under-reporting factor to be roughly the same across the chosen countries. All other parameters are left unchanged. Specifically, we leave the capacities of the health systems at the estimated values for Germany, as lacking TTI data and varying definitions of ICU treatment make any comparison difficult. We repeated the analysis presented above (Fig 4) for Finland, Italy and the Czech Republic (see Fig 5A–5D). Germany, Finland, and Italy would need a similarly high vaccine uptake in the population to prevent another severe wave. In the Czech Republic, a much smaller uptake is sufficient. The largest deviations in the necessary vaccine uptake are due to the initial seroprevalence, which we estimate to range from 5% in Finland to 30% in the Czech Republic. In contrast, the differences in age demographics and contact structures only have a minor effect on the dynamics (see also S1 Fig).

If no further measures remain in place to reduce the potential contagious contacts in school settings, the young age group (0–19 years) will drive infections after completing the vaccination program as they remain mostly unvaccinated. The combination of intense contacts and high susceptibility among school-aged children considerably increase the vaccine uptake required in the adult population to restrain a further wave (Fig 5E–5H). High seroprevalence, also in this age group, reduces the severity of this effect for the Czech Republic (Fig 5H).

## Discussion

Our results demonstrate that the pace of vaccination first and foremost determines the expected gain in freedom (i.e., lifting of restrictions) during and after completion of the

COVID-19 vaccination programs. Any premature lifting of restrictions risks another wave with high COVID-19 incidence and full ICUs. Moreover, the increase in freedom gained by these premature strategies is only transient because once ICU capacity is reached again, restrictions would have to be reinstated. Simultaneously, these early relaxations significantly increase morbidity and mortality rates, as a fraction of the population has not yet been vaccinated and thus remains susceptible. In contrast, maintaining low case numbers avoids another wave, and *still* allows to lift restrictions steadily and at a similar pace as with high case numbers. Despite this qualitative behavior being general, the precise quantitative results depend on several parameters and assumptions, which we discuss in the following.

The specific time evolution of the lifting of restrictions is dependent on the progress of the vaccination program. Therefore, a steady lifting of restrictions may start in May 2021, when the vaccination rate in the European Union gains speed. However, if the vaccination roll-out stalls more than we assume, the lifting of restrictions has to be delayed proportionally. In such a slowdown, the total number of cases and deaths until the end of the vaccination period increases accordingly. Thus, cautious lifting of restrictions and a fast vaccination delivery is essential to reduce death tolls and promptly increase freedom.

The spreading dynamics after concluding vaccination campaigns (Fig 4B and 4C) will be mainly determined by i) final vaccine uptake, ii) the contact network structure, iii) vaccine effectiveness, and iv) initial seroprevalence. Regarding vaccine uptake, we assumed that after the vaccination of every willing person, no further people would get vaccinated. This assumption enables us to study the effects of each parameter separately. However, vaccination willingness might change over time: it will probably be higher if reported case numbers and deaths are high, and vice versa. This poses a fundamental challenge: If low case numbers are maintained during the vaccine roll-out, the overall uptake might be comparably low, thus leading to a more severe wave once everyone has received a vaccination offer and restrictions are fully lifted. In contrast, a severe wave during vaccine roll-out might either increase vaccine uptake, because of individuals looking to protect themselves, or reduce it, because of damaged credibility on vaccine efficacy among vaccine hesitant groups. Thus, to avoid any further wave, policymakers have to maintain low case numbers *and* foster high vaccine uptake.

Besides vaccine uptake, the population's contact network also determines whether population immunity will be reached. We studied different real-world and theoretical possibilities for the contact matrices in Germany and other EU countries and evaluated how our results depend on the connectivity among age groups. For the long-term success of the vaccination programs, there must be exceptionally sensible planning of measures to prevent contagion among school-aged children. Otherwise, they could become the drivers of a novel wave because they might remain mostly unvaccinated. Provided adequate vaccine uptake among the adult population, our results suggest that reducing either the intensity of contacts or the infectiousness in that age group by half would be sufficient for preventing a rebound wave. This reduction is attainable by implementing soft-distancing measures, plus systematic, preventive random screening with regular COVID-19 rapid tests in school settings or via vaccination [22]. Although at the time of writing some vaccines have been provisionally approved for use in children aged 12–15 years old, vaccine uptake among children remains highly uncertain because of their very low risk for severe illness from COVID-19. We therefore did not include their vaccination in our model.

One of the largest uncertainties regarding the dynamics after vaccine roll-out arises from the efficacies of the vaccines. First, the sterilizing immunity effect (i.e., blocking the transmission of the virus), is still not well quantified and understood [24]. Second, the emergence of new viral variants that at least partially escape immune response is continuously under

investigation [35, 37, 38]. Furthermore, there is no certainty about whether escape variants might produce a more severe course of COVID-19 or whether reinfections with novel variants of SARS-CoV-2 would be milder. Therefore, we cannot conclusively quantify the level of contact reductions necessary in the long term to avoid a further wave of infections or whether such wave would overwhelm ICUs. However, for our default parameters, moderate contact reductions and hygiene measures would be sufficient to prevent further waves.

Although most examples are presented for countries from the European Union, our results can also be generalized to other countries. Differences across countries come from i) demographics, ii) varying seroprevalence —which originated from large differences in the severity of past waves—, iii) vaccines (types, availability, delivery scheme, and uptake), as well as iv) capacities of the health systems, including hospitals and TTI capabilities. For the EU, we find that during the mass vaccination phase, all these differences have only a minor effect on the pace at which restrictions can be lifted (cf. S1 Fig). However, differences become evident in the long term when most restrictions are lifted by the end of the vaccination campaigns. Demographics and contact patterns are qualitatively very similar across EU countries and thus do not strongly change the expected outcome. On the contrary, we found the initial seroprevalence to significantly determine the minimum vaccine uptake required to guard against further waves after the vaccine roll-out (cf. Fig 5). Naturally acquired immunity, like vaccinations, contributes to reducing the overall susceptibility of the population and thus impedes viral spread. Notably, naturally acquired immunity can compensate for drops in vaccine uptake in specific age groups unwilling to vaccinate or that cannot access the vaccine, e.g. in children. Furthermore, expected vaccine uptake considerably varies across EU countries (e.g., Serbia 38%, Croatia 41%, France 44%, Italy 70%, Finland 81% [6], Czech Republic 40% [39], Germany 80% [26]). The risk of rebound waves after the mass vaccinations might thus be highly heterogeneous across the EU.

Since we neither know what kind of escape variants might still surface nor their potential impact on vaccine efficacies or viral spread, maintaining low case numbers is the safest strategy for long-term planning. This strategy i) prevents avoidable deaths during vaccine roll-out, ii) offers better preparedness should escape variants emerge, and iii) lowers the risk of further waves because local outbreaks are easier to contain with efficient TTI. Hence, low case numbers only have advantages for health, society, and the economy. Furthermore, a low case number strategy would greatly profit from an EU-wide commitment, and coordination [15]. Otherwise, strict border controls with testing and quarantine policies need to be installed as drastically different case numbers between neighboring countries or regions promote destabilization; infections could (and will) propagate between countries triggering a "ping-pong" effect, especially if restrictions are not jointly planned. Therefore, promoting a high vaccine uptake and low case numbers strategy should not only be a priority for each country but also for the whole European community.

In practice, there are several ways to lower case numbers to the capacity limit of TTI programs without the need to enact stringent NPIs immediately. For example, if restrictions are lifted gradually but marginally slower than the rate vaccination pace would allow, case numbers will still decline. Alternatively, restrictions could be relaxed initially to an intermediate level where case numbers do not grow exponentially while giving people some freedom. In such circumstances one can take advantage of the reduced susceptibility to drive case numbers down without the need of stringent NPIs (S5(E)–S5(H) Fig).

To conclude, the opportunity granted by the progressing vaccination should not only be used to lift restrictions carefully but also to bring case numbers down. This will significantly reduce fatalities, allow to lift all major restrictions gradually moving into summer 2021, and guard against newly-emerging variants or potential further waves in the EU.

## Methods

### Model overview

We model the spreading dynamics of SARS-CoV-2 following a SEIRD-ICU deterministic formalism through a system of delay differential equations. Our model incorporates age-stratified dynamics, ICU stays, and the roll-out of a 2-dose vaccine. For a graphical representation of the infection and core dynamics, see Fig 6. The contagion dynamics include the effect of externally

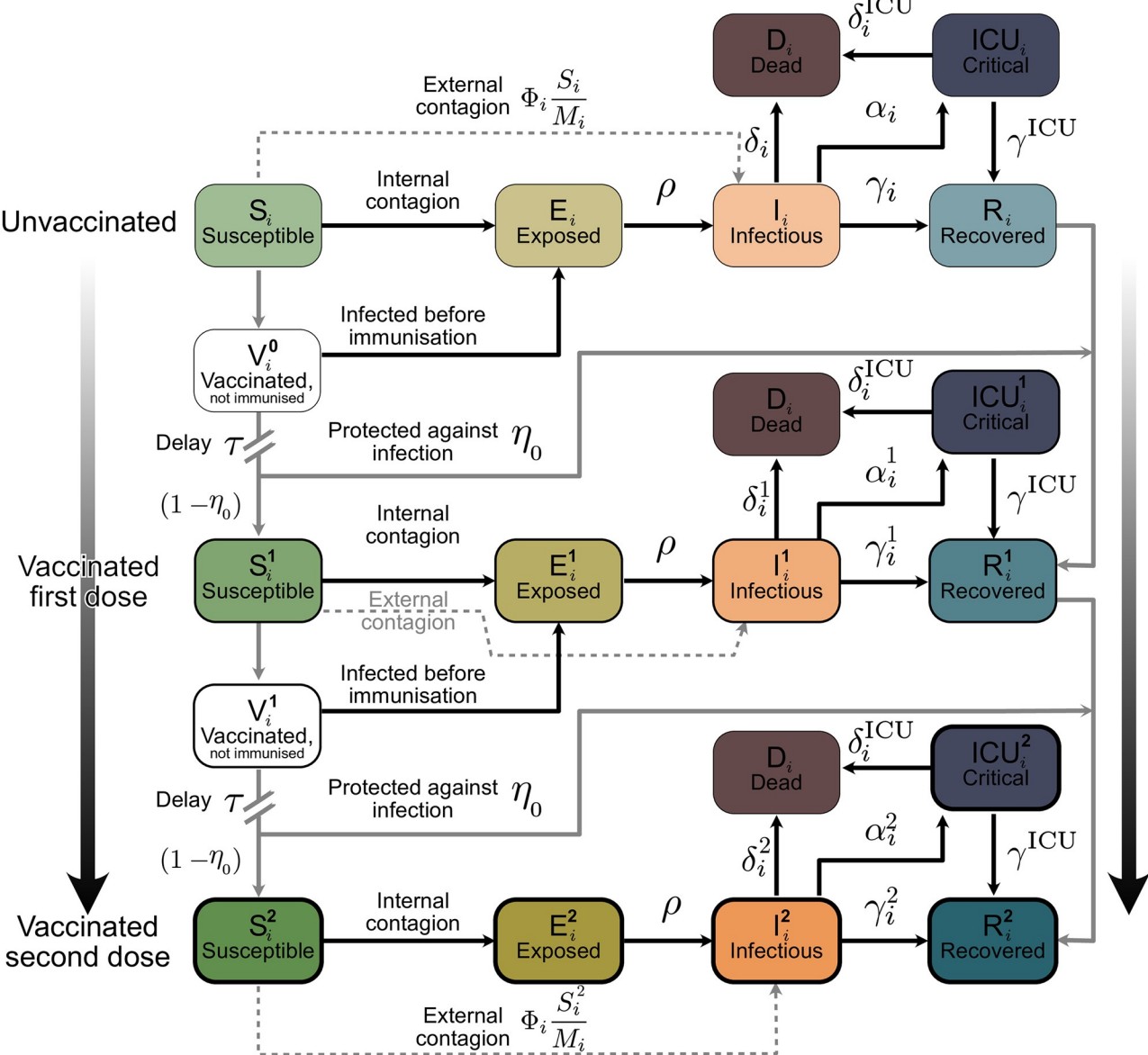

**Fig 6. Scheme of our age-stratified SEIRD-ICU+vaccination model.** The solid blocks in the diagram represent different SEIRD compartments. Solid black lines represent transition rates of the natural progression of the infection (contagion, latent period, and recovery). On the other hand, dashed lines account for external factors and vaccination. Solid gray lines represent non-linear transfers of individuals between compartments, e. g. through scheduled vaccination. From top to bottom, we describe the progression from unvaccinated to vaccinated, with stronger color and thicker edges indicating more protection from the virus. Subscripts $i$ indicate the age groups, while superscripts represent the number of vaccine doses that have successfully strengthened immune response in individuals receiving them. Contagion can occur internally, where an individual from age group $i$ can get infected from an infected person from any age group, or externally, e. g., abroad on vacation. If the contagion happens externally, we assume that the latent period is already over when the infected returns and, hence, they are immediately put into the infectious compartments $I_i^v$.

acquired infections as a non-zero influx $\Phi_i$ based on the formalism previously developed by our group [18, 19]: susceptible individuals of a given age group $i$ ($S_i$) can acquire the virus from infected individuals from any other age group $j$ and subsequently progress to the exposed ($S_i \rightarrow E_i$) and infectious ($E_i \rightarrow I_i$) compartments. They can also acquire the virus externally. However, in this case, they progress directly to the infectious compartment ($S_i \rightarrow I_i$), i.e., they get infected abroad, and by the time they return, the latent period is already over. Individuals exposed to the virus ($E_i$) become infectious after the latent period and thus progress from the exposed to the infectious compartments ($I_i$) at a rate $\rho$ ($E_i \rightarrow I_i$). The infectious compartment has three different possible transitions: i) direct recovery ($I_i \rightarrow R_i$), ii) progression to ICU ($I_i \rightarrow$ ICU$_i$) or iii) direct death ($I_i \rightarrow D_i$). Individuals receiving ICU treatment can either recover (ICU$_i \rightarrow R_i$) or decease (ICU$_i \rightarrow D_i$).

A contact matrix weights the infection probability between age groups. We investigated three different settings for the contact structure to assess its impact on the spreading dynamics of COVID-19: i) Interactions between age groups are proportional to the group size, i.e., the whole population is mixed perfectly homogeneously, ii) interactions are proportional to pre-COVID contact patterns in the EU population [28], and iii) interactions are proportional to "almost" pre-COVID contact patterns [28], i.e., the contact intensity in the youngest age group (0–19 years) is halved. This accounts for some preventive measures kept in place in schools, e.g., regular rapid testing or smaller class sizes. Scenario iii) is the default scenario unless explicitly stated. However, figures for Scenarios i) and ii) are provided in S9–S14 Figs. We scale all the contact structures by a linear factor, which increases or decreases the stringency of NPIs so that the settings are comparable. However, the scaling above does not account for heterogeneous NPIs acting only on contacts between specific age groups, such as workplace or school restrictions.

Our model includes the effect of vaccination, where vaccines are administered with an age-stratified two-dosage delivery scheme. The scheme does not discriminate on serological status, i.e., recovered individuals with natural antibodies may also access the vaccine when offered to them. Immunization, understood as the development of proper antibodies against SARS-CoV-2, does not occur immediately after receiving the vaccination dose. Thus, newly vaccinated individuals get temporarily put into extra compartments ($V_i^0$ and $V_i^1$ for the first and second dose respectively) where, if infected, they would progress through the disease stages as if they would not have received that dose. For modeling purposes, we assume that a sufficient immune response is build up $\tau$ days after being vaccinated ($V_i^0 \rightarrow S_i^1$ and $V_i^1 \rightarrow S_i^2$), and that a fraction $p_i(t)$ of those individuals that received the dose acquire the infection before being immunized. Furthermore, there is some evidence that the vaccines partially prevent the infection with and transmission of the disease [40, 41]. Our model incorporates the effectiveness against infection following an 'all-or-nothing' scheme, removing a fraction of those vaccinated individuals to the recovered compartments ($V_i^0 \rightarrow R_i^1$ and $V_i^1 \rightarrow R_i^2$), thus assuming that they would not participate in the spreading dynamics. However, we consider those vaccinated individuals with a breakthrough infection have a lower probability of going to ICU or to die than unvaccinated individuals, i.e., effectiveness against severe disease follows a 'leaky' scheme. Furthermore, we assume those individuals carry a lower viral load and thus are less infectious by a factor of two [25]. All parameters and values are listed in Table 2.

We model the mean-field interactions between compartments by transition rates, determining the timescales involved. These transition rates can implicitly incorporate both the time course of the disease and the delays inherent to the case-reporting process. In the different scenarios analyzed, we include a non-zero influx $\Phi_i$, i.e., new cases that acquired the virus from outside. Even though this influx makes a complete eradication of SARS-CoV-2 impossible,

**Table 2. Model parameters.** The range column either describes the range of values used in the various scenarios, or if values depend on the age group (indexed by *i*), the lowest and highest value across age-groups.

| Parameter | Meaning | Value (default) | Range | Units | Source |
|---|---|---|---|---|---|
| $R_t$ | Reproduction number (gross) | 1.00 | 0–3.5 | — | Assumed |
| $\eta$ | Vaccine protection against transmission | 0.75 | 0.5–0.85 | — | [24, 40, 41] |
| $\kappa$ | Vaccine efficacy (against severe disease) | 0.9 | 0.7–0.95 | — | [23, 57] |
| $\sigma^v$ | Relative virulence of unvaccinated and vaccinated individuals | [1.0, 0.5, 0.5] | 0.5–1 | — | [25] |
| $\tau$ | Immunization delay | 7 | — | days | [24, 31] |
| $v_r$ | Random vaccination fraction | 0.35 | — | — | [64, 65] |
| $M_i$ | Population group size | Table 4 | — | people | [43] |
| $u_i$ | Vaccine uptake | Table 4 | — | — | [6] |
| $\rho$ | Transition rate $E \rightarrow I$ | 0.25 | — | day$^{-1}$ | [66, 67] |
| $\gamma_i^v$ | Recovery rate from $I_i^v$ | Table 5 | 0.088–0.1 | day$^{-1}$ | [54–56] |
| $\gamma_i^{\text{ICU}}$ | Recovery rate from $\text{ICU}_i^v$ | Table 5 | 0.08–0.2 | day$^{-1}$ | [50, 52, 68] |
| $\delta_i^v$ | Death rate from $I_i^v$ | Table 5 | $10^{-6}$–0.005 | day$^{-1}$ | [50, 52, 68] |
| $\delta_i^{\text{ICU}}$ | Death rate from $\text{ICU}_i^v$ | Table 5 | 0.0055–0.083 | day$^{-1}$ | [50, 52, 68] |
| $\alpha_i^v$ | Transition rate $I \rightarrow \text{ICU}$ | Table 5 | $10^{-5}$–0.007 | day$^{-1}$ | [50, 52, 68] |
| $\Phi_i$ | Infections from external sources | 1 | — | cases day$^{-1}$ per million | Assumed |
| $p_i(t)$ | Fraction of individuals getting infected before acquiring antibodies | — | — | — | Eq (34) |
| $\bar{\gamma}$ | Effective removal rate from infectious compartment | — | — | day$^{-1}$ | $(\gamma_i^v + \alpha_i^v + \delta_i^v)$ |
| $f_i^1(t), f_i^2(t)$ | Administered 1$^{\text{st}}$ and 2$^{\text{nd}}$ vaccine doses | — | — | doses/day | Eqs (19) and (20) |

different outcomes in the spreading dynamics might arise depending on both contact intensity and TTI [18]. Additionally, we include the effects of non-compliance and unwillingness to be vaccinated as well as the effects of the TTI capacities from health authorities, building on [19]. Throughout the manuscript, we do not make explicit differences between symptomatic and asymptomatic infections. However, we implicitly consider asymptomatic infections by accounting for their effect on modifying the reproduction number $R_t$ and all other epidemiological parameters. To assess the lifting of restrictions in light of progressing vaccinations, we use a Proportional-Derivative (PD) control approach to adapt the internal reproduction number $R_t$ targeting controlled case numbers or ICU occupancy.

## Model equations

The contributions of the spreading dynamics and the age-stratified vaccination strategies are summarized in the equations below. They govern the infection dynamics between the different age groups, each of which is represented by their susceptible-exposed-infectious-recovered-dead-ICU (SEIRD+ICU) compartments for all three vaccination statuses. We assume a regime that best resembles the situation in Germany at the beginning of March 2021, and we estimate the initial conditions for the different compartments of each age group accordingly. Furthermore, we assume that neither post-infection immunity [42] nor the immunization obtained through the different dosages of the vaccine vanish significantly in the considered time frames. The spreading parameters completely determine the resulting dynamics (characterized by the different age- and dose-dependent parameters, together with the gross reproduction number $R_t$) and the vaccination logistics.

All of the following parameters and compartments are shortly described in Tables 2 and 3. Some of these are elaborated in more detail in the following sections. Subscripts *i* in the equations denote the different age groups, while superscripts denote the vaccination status:

**Table 3. Model variables.** Subscripts $i$ denote the $i$th age group, superscripts the vaccination status (unvaccinated, immunized by one dose, by two doses).

| Variable | Meaning | Units | Explanation |
|---|---|---|---|
| $S_i$, $S_i^1$, $S_i^2$ | Susceptible pools | people | Non-infected people that may acquire the virus. |
| $V_i^0$, $V_i^1$ | Vaccinated pools | people | Non-infected people that have been vaccinated but have not developed antibodies yet, thus may acquire the virus. |
| $E_i$, $E_i^1$, $E_i^2$ | Exposed pools | people | Infected people in latent period. Cannot spread the virus. |
| $I_i$, $I_i^1$, $I_i^2$ | Infectious pools | people | Currently infectious people. |
| $ICU_i$, $ICU_i^1$, $ICU_i^2$ | ICU pools | people | Infected people receiving ICU treatment, isolated. |
| $D_i$, $D_i^1$, $D_i^2$ | Dead pools | people | Dead people. |
| $R_i$, $R_i^1$, $R_i^2$ | Recovered pools | people | Recovered/immune people that have acquired post-infection or sterilizing vaccination immunity. |
| $\hat{N}^{obs}$ | Observed new infections | people day$^{-1}$ | Daily new infections, including reporting delays. Eq (42) |
| $\hat{R}_t^{obs}$ | Observed reproduction number | – | The reproduction number that can be estimated only from the observed cases: $\hat{R}_t^{obs} = \hat{N}^{obs}(t)/\hat{N}^{obs}(t - 4)$. |

unvaccinated ($^0$ or none), immunized by one dose ($^1$), or by two doses ($^2$).

$$\frac{dS_i}{dt} = \underbrace{-\bar{\gamma} R_t S_i \sum_{j,v} C_{ji} \frac{\sigma^v I_j^v}{M_j}}_{\text{internal contagion}} - \underbrace{f_i^1(t)\frac{S_i}{S_i + R_i}}_{\text{administering first dose}} - \underbrace{\frac{S_i}{M_i}\Phi_i}_{\text{external contagion}} \tag{1}$$

$$\frac{dV_i^0}{dt} = \underbrace{-\bar{\gamma} R_t V_i^0 \sum_{j,v} C_{ji} \frac{\sigma^v I_j^v}{M_j}}_{\text{internal contagion}} + \underbrace{f_i^1(t)\frac{S_i}{S_i + R_i}}_{\text{administering first dose}} \cdots$$
$$\cdots - \underbrace{f_i^1(t-\tau)\frac{S_i}{S_i + R_i}\Big|_{t-\tau}(1 - p_i(t))}_{\text{first dose showing effect}} - \underbrace{\frac{V_i^0}{M_i}\Phi_i}_{\text{external contagion}} \tag{2}$$

$$\frac{dS_i^1}{dt} = \underbrace{-\bar{\gamma} R_t S_i^1 \sum_{j,v} C_{ji} \frac{\sigma^v I_j^v}{M_j}}_{\text{internal contagion}} - \underbrace{f_i^2(t)\frac{S_i^1}{S_i^1 + R_i^1}}_{\text{administering second dose}} \cdots$$
$$\cdots + \underbrace{(1 - \eta_0)f_i^1(t-\tau)\frac{S_i}{S_i + R_i}\Big|_{t-\tau}(1 - p_i(t))}_{\text{first dose (not immune)}} - \underbrace{\frac{S_i^1}{M_i}\Phi_i}_{\text{external contagion}} \tag{3}$$

$$\frac{dV_i^1}{dt} = \underbrace{-\bar{\gamma} R_t V_i^1 \sum_{j,v} C_{ji} \frac{\sigma^v I_j^v}{M_j}}_{\text{internal contagion}} + \underbrace{f_i^2(t)\frac{S_i^1}{S_i^1 + R_i^1}}_{\text{administering second dose}} \cdots$$
$$\cdots - \underbrace{f_i^2(t-\tau)\frac{S_i^1}{S_i^1 + R_i^1}\Big|_{t-\tau}(1 - p_i(t))}_{\text{second dose showing effect}} - \underbrace{\frac{V_i^1}{M_i}\Phi_i}_{\text{external contagion}} \tag{4}$$

$$\frac{dS_i^2}{dt} = \underbrace{-\bar{\gamma}R_t S_i^2 \sum_{j,v} C_{ji} \frac{\sigma^v I_j^v}{M_j}}_{\text{internal contagion}} + \underbrace{(1-\eta_0)f_i^2(t-\tau)\frac{S_i^1}{S_i^1+R_i^1}\bigg|_{t-\tau}(1-p_i(t))}_{\text{second dose (not immune)}} \cdots$$

$$\cdots - \underbrace{\frac{S_i^2}{M_i}\Phi_i}_{\text{external contagion}}$$

(5)

$$\frac{dE_i}{dt} = \underbrace{\bar{\gamma}R_t\left(S_i+V_i^0\right)\sum_{j,v} C_{ji}\frac{\sigma^v I_j^v}{M_j}}_{\text{internal contagion}} - \underbrace{\rho E_i}_{\text{end of latency}}$$

(6)

$$\frac{dE_i^1}{dt} = \underbrace{\bar{\gamma}R_t\left(S_i^1+V_i^1\right)\sum_{j,v} C_{ji}\frac{\sigma^v I_j^v}{M_j}}_{\text{internal contagion}} - \underbrace{\rho E_i^1}_{\text{end of latency}}$$

(7)

$$\frac{dE_i^2}{dt} = \underbrace{\bar{\gamma}R_t S_i^2 \sum_{j,v} C_{ji}\frac{\sigma^v I_j^v}{M_j}}_{\text{internal contagion}} - \underbrace{\rho E_i^2}_{\text{end of latency}}$$

(8)

$$\frac{dI_i}{dt} = \underbrace{\rho E_i}_{\text{end of latency}} - \underbrace{\bar{\gamma}I_i}_{\text{recovery, ICU admission, or death}} + \underbrace{\frac{S_i+V_i^0}{M_i}\Phi_i}_{\text{external contagion}}$$

(9)

$$\frac{dI_i^1}{dt} = \underbrace{\rho E_i^1}_{\text{end of latency}} - \underbrace{\bar{\gamma}I_i^1}_{\text{recovery, ICU admission, or death}} + \underbrace{\frac{S_i^1+V_i^1}{M_i}\Phi_i}_{\text{external contagion}}$$

(10)

$$\frac{dI_i^2}{dt} = \underbrace{\rho E_i^2}_{\text{end of latency}} - \underbrace{\bar{\gamma}I_i^2}_{\text{recovery, ICU admission, or death}} + \underbrace{\frac{S_i^2}{M_i}\Phi_i}_{\text{external contagion}}$$

(11)

$$\frac{d\text{ICU}_i^v}{dt} = -\underbrace{(\delta_i^{\text{ICU}}+\gamma_i^{\text{ICU}})\text{ICU}^v}_{\text{recovery or death}} + \underbrace{\alpha_i^v I_i^v}_{\text{ICU admission}}$$

(12)

$$\frac{dD_i}{dt} = \underbrace{\sum_v (\delta_i^{\text{ICU}}\text{ICU}_i^v + \delta_i^v I_i^v)}_{\text{total deaths}}$$

(13)

$$\frac{dR_i}{dt} = \underbrace{\gamma_i^{\text{ICU}}\text{ICU}_i + \gamma_i I_i}_{\text{recovery}} - \underbrace{f_i^1(t)\frac{R_i}{S_i+R_i}}_{\text{first dose}}$$

(14)

$$\frac{dR_i^1}{dt} = \underbrace{\gamma_i^{\text{ICU}}\text{ICU}_i^1 + \gamma_i^1 I_i^1}_{\text{recovery}} + \underbrace{f_i^1(t)\frac{R_i}{S_i+R_i}}_{\text{first dose after recovery}} - \underbrace{f_i^2(t)\frac{R_i^1}{S_i^1+R_i^1}}_{\text{second dose}} \cdots$$

$$\cdots + \underbrace{\eta_0 f_i^1(t-\tau)\frac{S_i}{S_i+R_i}\Big|_{t-\tau}(1-p_i(t))}_{\text{first dose (sterilizing immunity)}}$$

(15)

$$\frac{dR_i^2}{dt} = \underbrace{\gamma_i^{\text{ICU}}\text{ICU}_i^2 + \gamma_i^2 I_i^2}_{\text{recovery}} + \underbrace{f_i^2(t)\frac{R_i^1}{S_i^1+R_i^1}}_{\text{second dose after recovery}} \cdots$$

$$\cdots + \underbrace{\eta_0 f_i^2(t-\tau)\frac{S_i^1}{S_i^1+R_i^1}\Big|_{t-\tau}(1-p_i(t))}_{\text{second dose (sterilizing immunity)}}$$

(16)

## Contact structure and the effect of NPIs on the contact levels

We model the probability of a susceptible individual from age group $i$ to get infected from an individual from age group $j$ to be proportional to the –effective– incidence in that group $(\sum_v I_j^v \sigma^v)$ and the contact intensity between the two groups, given by the entries $(C)_{ij}$ of a contact matrix $C$ scaled with the gross reproduction number $R_t$. The contact matrices are normalized to force their largest eigenvalue (i.e., their spectral radius) to be 1, so that, when multiplied with $R_t$, their spectral radius equals $R_t$. The total contact levels for different levels of NPIs are then just linearly scaled with $R_t$. We thus neglect any inhomogeneities in the NPIs that might affect contact between specific age groups more than others.

As described previously, we study three different configurations for the contact matrix $C$: i) a perfectly homogeneously mixed population, ii) pre-COVID structure in the EU population [28], and iii) "almost" pre-COVID contact structure [28], but with reduced potentially-contagious contacts in the youngest age group (0–19 years) accounting for some preventive measures kept in place in schools. If not explicitly stated otherwise, the default contact matrix we use in the main text is always the intermediate "almost" pre-COVID contact structure matrix. For the three scenarios, we analyze the demographics and contact structures in Germany, Finland, the Czech Republic, and Italy as a sample for varying demographics across the EU.

**First scenario: Homogeneous contact structure.** In this scenario, we consider that everyone has the same probability of meeting anyone from any other age group. The probability of meeting somebody from a given age group is thus proportional to the fraction of this age group within the whole population. Let $f$ be the column vector collecting these fractions, $f_i = \frac{M_i}{M}$, the contact matrix for the $n$ age-groups herein considered $C \in \mathbb{R}^{n\times n}$ is thus given by

$$(C)_{ij} = f_j, \forall j \tag{17}$$

and can be seen in Fig 7A, 7D, 7G and 7J, for the chosen demographics. Note that by this construction the largest eigenvalue of this $C$ (i.e., its spectral radius) is automatically 1 for any demographics, i.e., for any $f$ that fulfills $\Sigma_j f_j = 1$ (proof in S1 Supplementary Note).

**Second scenario: Pre-COVID contact intensity, real-world contact structure.** Here, we use the whole contact matrices from before the pandemic reported with one-year age resolution in [28], converted into the age brackets that we chose. We normalize them by their

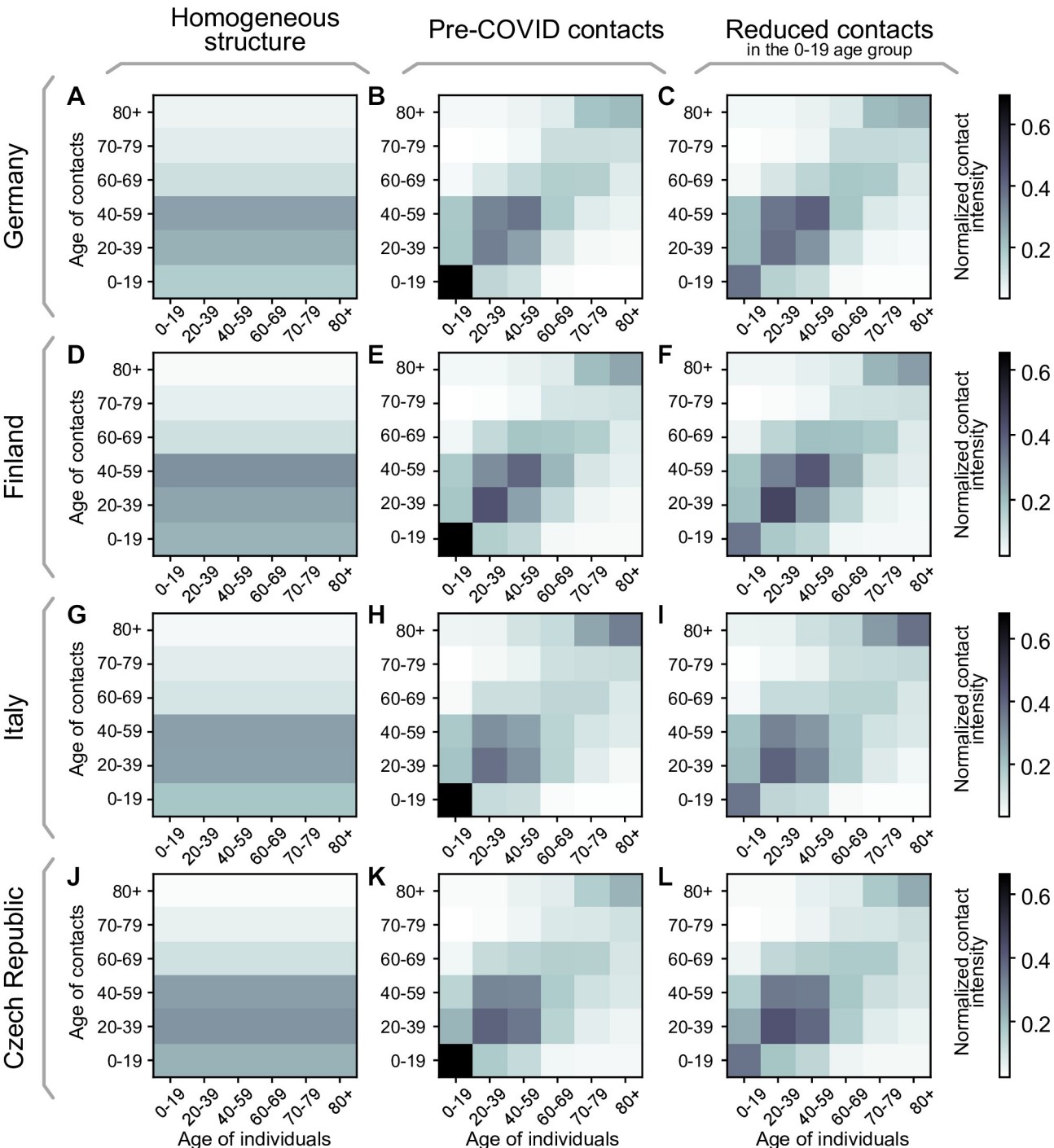

**Fig 7. Contact structures for different EU countries in the three scenarios.** The chosen contact matrices for i) homogeneous contact structure, ii) pre-COVID contact structure, and iii) "almost" pre-COVID structure with reduced potentially-contagious contacts in schools for Germany (**A-C**), Finland (**D-F**), Italy (**G-I**) and the Czech Republic (**J-L**). Entries of the matrices show the contact intensity between age groups normalized to give each matrix a spectral radius of 1.

spectral radius, leaving their internal contact structure intact. This scenario thus resembles completely homogeneous NPIs that affect every possible contact equally. The matrices are given in Fig Fig 7B, 7E, 7H and 7K for the chosen countries.

**Third scenario: "Almost" pre-COVID contact intensity, real-world contact structure.** Finally, we again use the contact matrices from before the pandemic reported in [28] but adapt them to reduce the intensity of contacts of the youngest age group by half, accounting for those measures that remain in place to prevent contagion and mitigate outbreaks in school settings. Specifically, we halve the matrix element connecting the 0–19 age group with itself and normalize the obtained contact matrix $C$ by its spectral radius. As can be seen in the resulting matrices, given in Fig 7C, 7F, 7I and 7L, this affects that the main contributions in the contacts are more evenly spread in the 0–59 year age groups. This serves as a first approximation to the contact structure with inhomogeneous NPIs targeting different age groups differently both in a complete lockdown, as well as some continued measures in schools.

## Vaccination dynamics and logistics

In real-world settings, not every person accepts the vaccine when offered. Additionally, vaccine uptake is bounded because some vulnerable groups cannot be vaccinated because of health-related reasons. A systematic survey [26] estimates the vaccine uptake to be approximately 80% across the adult population in Germany, which we choose as our baseline. Due to a higher perception of the risk caused by an infection, we expect that the uptake is higher for elderly population. Thus, we set the uptake $u_i$ to be age-group dependent. Besides the default 80%, we choose two more sets of uptakes averaging to a total of 70% and 90%, respectively. We suppose that an increase in the uptake is possible by education and information measures. They are listed in Table 4. We linearly interpolate between the three values to model arbitrary total vaccine uptakes.

Using official data of the German vaccine stock and stock projections [44, 45] we build up an estimated delivery function $w_T$ that models the weekly number of doses delivered as a function of time. We assume it takes a logistic form, as we assume the number of daily doses increases strongly at the beginning until it reaches a stable level. Adapting the logistic function to the German stock projection (see Fig 8) yields:

$$w_T(week) = \frac{11 \times 10^6 \text{ doses}}{1 + \exp\left(-0.17(week - 21)\right)}, \tag{18}$$

where the parameters were chosen to roughly match past and projected deliveries, taking into account that some delays in the projections might appear because of logistic or manufacturing issues. Since the vaccine deliveries and distributions are done collectively and uniformly in the

**Table 4. Parameters for the three main different vaccine uptake scenarios for Germany.** The averages are to be understood across the vaccinable (16+) population. Slightly rescaled uptakes for Finnish, Italian and Czech age-demographics can be found in S1, S2 and S3 Tables.

| Group ID | age group | eligible fraction | minimal uptake $u_i$ | mid uptake $u_i$ (default) | maximal uptake $u_i$ | population fraction [43] $M_i/M$ |
|---|---|---|---|---|---|---|
| 1 | 0–19 | 0.2 (16+) | 0.58 | 0.73 | 0.88 | 0.18 |
| 2 | 20–39 | 1.0 | 0.64 | 0.76 | 0.89 | 0.25 |
| 3 | 40–59 | 1.0 | 0.69 | 0.79 | 0.90 | 0.28 |
| 4 | 60–69 | 1.0 | 0.74 | 0.82 | 0.91 | 0.13 |
| 5 | 70–79 | 1.0 | 0.79 | 0.86 | 0.92 | 0.09 |
| 6 | >80 | 1.0 | 0.85 | 0.89 | 0.93 | 0.07 |
| average | — | — | 0.70 | 0.80 | 0.90 | — |

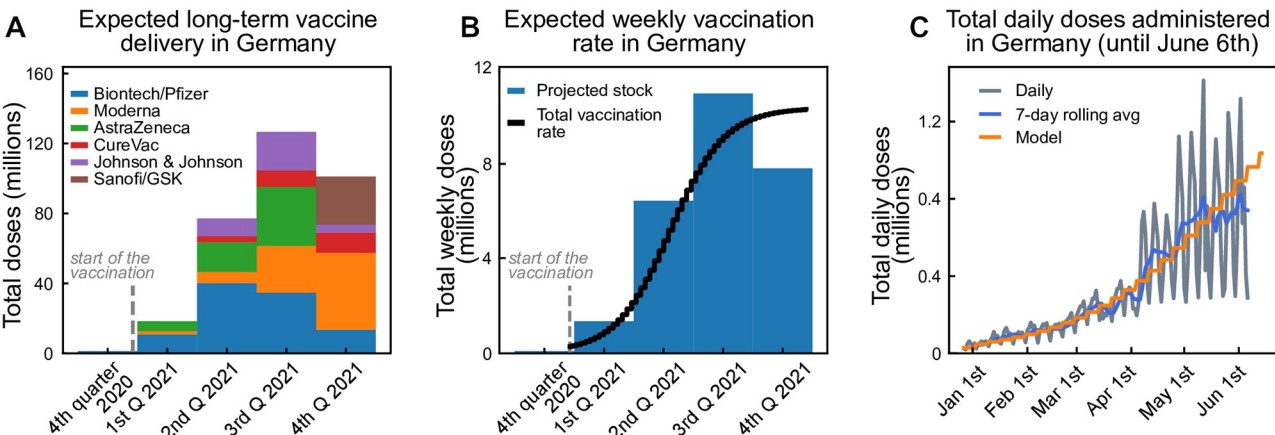

**Fig 8. Estimated vaccination rates for Germany.** From the announced vaccination stock, we estimate the vaccination delivery function. **A**: Total aggregated doses of different vaccine producers in Germany. **B**: Equivalent amount of 2-dose vaccines available per week in Germany, parameterized using a logistic function. **C**: Comparison between expected and observed vaccination progress in Germany.

EU, we scale this German projection by the respective population sizes for the other countries studied herein (Finland, Italy, Czech Republic). We further assume that because of logistic delays, the vaccination of the delivered doses occurs with some delay, which we model as a convolution with an empirical delay kernel given by $K = [0.6, 0.3, 0.1]$ (fraction of vaccines administered in the same, second and third week following delivery). With that, we get the total vaccination rates per week.

These doses are distributed among the age groups, taking into account that each individual requires two doses, spaced by at least four weeks, aware of the potential benefits of further delaying the two doses [46].

The vaccine prioritization order is the following:

1. First, to meet the demand of second doses, $\tau_{vac}$ weeks after the first dose.

2. Second, to distribute a fraction $v_r$ of the remaining doses uniformly among age groups, to model the earlier vaccination of exposed occupations (health sector, first responders, among others).

3. Last, to plan the rest of the doses for the oldest age group that has not been fully vaccinated yet.

Exceptions to rule 3 are the low-risk groups 16–19, 20–39, and 40–59 that get vaccinated simultaneously. For each age group, only a fraction $u_i$ is vaccinated because of limited willingness to get vaccinated (Table 4). In addition, the total number of vaccinations in the youngest age group 0–19 is further reduced since we consider only a fraction of around 20% (fraction of 16–19 year-old individuals in the group) to be *eligible* for vaccination (see Table 4). The uptake $u_i$ in this age group is thus understood only among the eligible individuals.

This procedure results in the number of first $w_i^1(week)$ and second doses $w_i^2(week)$ vaccinated to the age group $i$ as a function of the week. Dividing by 7 we obtain the daily administered first and second doses for age group $i$

$$f_i^1(t) = w_i^1(\lfloor t/7 \rfloor)/7 \quad \text{and} \tag{19}$$

$$f_i^2(t) = w_i^2(\lfloor t/7 \rfloor)/7. \tag{20}$$

**Table 5. Age-dependent parameters.**

| Age group | ICU admission rate $\alpha_i$ (days$^{-1}$) | Death rate in I $\delta_i$ (days$^{-1}$) | Natural recovery rate $\gamma_i$ (days$^{-1}$) | Death rate in ICU $\delta_i^{\text{ICU}}$ (days$^{-1}$) | ICU recovery rate $\gamma_i^{\text{ICU}}$ (days$^{-1}$) | Avg. duration in ICU $T_{\text{res}}^{\text{ICU}}$ (days) |
|---|---|---|---|---|---|---|
| 0–19 | 0.000014 | 0.000002 | 0.09998 | 0.005560 | 0.194440 | 5 |
| 20–39 | 0.000204 | 0.000014 | 0.09978 | 0.007780 | 0.192220 | 5 |
| 40–59 | 0.001217 | 0.000111 | 0.09867 | 0.006164 | 0.084745 | 11 |
| 60–69 | 0.004031 | 0.000317 | 0.09565 | 0.009508 | 0.081401 | 11 |
| 70–79 | 0.005435 | 0.001422 | 0.09314 | 0.019756 | 0.091355 | 9 |
| >80 | 0.007163 | 0.004749 | 0.08809 | 0.082433 | 0.084233 | 6 |

### Age-stratified transition rates

Here, we will introduce the transition rates used in the model equations; details about their estimation are presented in the later sections.

The recovery rate $\gamma_i$ of a given age group describes the recovery without the need for critical care. It is estimated from the literature. We expect this parameter to vary across age groups, mainly because of the strong correlation between the severity of symptoms and age. Age-resolved recovery rates estimated from data of the non-vaccinated population in Germany are listed in Table 5.

The ICU recovery rate $\gamma_i^{\text{ICU}}$ is the rate of a given age group for leaving ICU care. This parameter varies across age groups, mainly because of the strong correlation between the severity of symptoms, age, and duration of ICU stay. Age-resolved ICU recovery rates estimated from data of the non-vaccinated population in Germany are listed in Table 5.

The ICU admission rate $\alpha_i$ of a given age group describes the transition from the infected compartment to the ICU compartment. It accounts for those cases developing symptoms where intensive care is required and is estimated from the literature. We expect this parameter to vary across age groups, mainly because of the strong correlation between the severity of symptoms and age. Age-resolved ICU-transition rates estimated from data of the non-vaccinated population in Germany are listed in Table 5. Further, we assume that anyone requiring intensive care would have access to ICU beds and care.

The death rate $\delta_i$ also varies across age groups, mainly because of the strong correlation between the severity of symptoms and age. This parameter accounts for those individuals dying because of COVID-19, but without being treated in the ICU. In that way, it is expected to be even smaller than the infection fatality ratio (IFR). Age-resolved death rates (outside ICU) estimated from data of the non-vaccinated population in Germany are listed in Table 5.

The death rate in ICU $\delta_i^{\text{ICU}}$ also varies across age groups, mainly because of the strong correlation between the severity of symptoms and age. In addition, this parameter accounts for those individuals dying because of COVID-19 when being treated in the ICU. In that way, it is expected to be even larger than the case fatality ratio CFR. Age-resolved ICU death rates estimated from data of the non-vaccinated population in Germany are listed in Table 5.

We estimate these age-dependent rates by combining hospitalization data with published IFR data. A comparison of ICU transition rates $\alpha_i^v$ across the EU is difficult as the definition of stationary treatment differs with regard to *hospitalization*, *ICU low* and *high-care*. In order to obtain sensible estimates for these rates, we need to consider the size of the unobserved pool in each age group. Our analysis of ICU transition rates is based on 14043 hospitalization reports collected in Germany between early 2020 and Oct. 26, 2020, as part of the official reporting data [47]. Those reports contain 20-year wide age strata but only represent a small sub-sample of all ICU-admissions ($n = 723$). A complete count of ICU-admissions is maintained by the

*Deutsche Interdisziplinäre Vereinigung für Intensiv- und Notfallmedizin* [48], without additional patient-data, like age. 19250 ICU admissions were reported throughout the same time frame. We estimated the number of ICU admissions in each 20-year wide age group by combining both sources, matching well with German studies on the first wave [49].

Throughout the first and second wave, the per age-group case-fatality rates (CFRs) in Germany are more than two times larger than the age-specific infection fatality rates (IFRs) estimated by [27, 50]. This difference indicates unobserved infections. Seroprevalence studies from Q3 2020 [51] confirm the existence of unobserved pools. The total number of infections in each age group is inferred from observed deaths assuming the age-specific IFR from [27]. $\alpha_i^v$ (*low-* and *high-care*) is calculated by dividing estimated ICU-admissions in each age group by the estimated total infections in each of those groups. A similar method is applied for the ICU-death-rate $\delta_i^{\text{ICU}}$ by taking hospitalization-deaths from [47] as a proxy for the age distribution.

The ICU-rates from the 10-year wide age-groups [52] based on French data (*high-care* only) were used to subdivide the 20-year wide age-group 60–79, replicating the French rate-ratio between 60–69 and 70–79 for the German ICU-ratios, while maintaining the German age-agnostic ICU-rate. Noteworthy, there is great variability between the reported ICU rates among different countries, and it seems to be more a problem of reporting criteria rather than differences in virus and host response [53]. Furthermore, as treatments become more effective compared to the first wave, the residence times have decreased in the second wave [30], thus modifying the transition rates.

We also considered the influence of our decision to use the IFR of O'Driscoll et al. [27] instead of Levin et al. [50]. The IFR from Levin et al. is about 50% larger and would lead to a lower level of infections overall in our scenarios, therefore reducing the fraction of natural immunity acquired at the end of the scenarios.

## Estimation of general transition rates

After listing all transition rates that we consider in our work, we will now explain how we estimate them. Since we have to start somewhere, let us look at the $\text{ICU}_i$ compartment first (see Fig 6 top right). The differential equation, without influx and including the initial condition $\text{ICU}_0$, is given by

$$\text{ICU}_i' = \underbrace{-\delta_i^{\text{ICU}}\text{ICU}_i}_{\text{to } D_i} - \underbrace{\gamma_i^{\text{ICU}}\text{ICU}_i}_{\text{to } R_i}, \qquad \text{ICU}_i(0) = \text{ICU}_0. \tag{21}$$

The solution of this ODE is known to be

$$\text{ICU}_i = \text{ICU}_0 \exp\left(-(\delta_i^{\text{ICU}} + \gamma_i^{\text{ICU}})t\right). \tag{22}$$

If we know the average $\text{ICU}_i$ residence time $T_{\text{res}}^{\text{ICU}}$, we can obtain an expression for $(\delta_i^{\text{ICU}} + \gamma_i^{\text{ICU}})$:

$$\delta_i^{\text{ICU}} + \gamma_i^{\text{ICU}} = \frac{1}{T_{\text{res}}^{\text{ICU}}}. \tag{23}$$

Further, assuming that a fraction $f_\delta$ of those individuals being admitted to ICUs would die, we obtain an expression linking all rates:

$$f_\delta = \frac{\text{\# people dead by } t = \infty}{\text{people entering ICU}_i \text{ at } t = 0} = \frac{\delta_i^{\text{ICU}} \cancel{\text{ICU}_0} \int_0^\infty \exp\left(-\frac{t}{T_{\text{res}}^{\text{ICU}}}\right) dt}{\cancel{\text{ICU}_0}} = \delta_i^{\text{ICU}} T_{\text{res}}^{\text{ICU}}. \tag{24}$$

Therefore, the transition rates are given by:

$$\delta_i^{\text{ICU}} = \frac{f_\delta}{T_{\text{res}}^{\text{ICU}}} \qquad \text{and} \qquad \gamma_i^{\text{ICU}} = \frac{(1 - f_\delta)}{T_{\text{res}}^{\text{ICU}}}. \tag{25}$$

Using this modeling approach, we implicitly assume the time scales at which people leave the ICU through recovery or death to be the same, i. e., the average ICU stay duration is independent of the outcome of the course of the disease.

Similarly, we can estimate the infected-to-death rate ($\delta_i$), the infected-to-ICU transition rate (ICU admission rate $\alpha_i$) and the infected-to-recovered rate ($\gamma_i$) based on these fractions and average times. If we assume that all the relevant median times are the same, we obtain the following expressions for the rates:

$$\delta_i = \frac{f_{I_i \to D_i}}{T_{\text{res}}^I}, \qquad \alpha_i = \frac{f_{\text{ICU}}}{T_{\text{res}}^I}, \qquad \gamma_i = \frac{(1 - (f_{I_i \to D_i} + f_{\text{ICU}}))}{T_{\text{res}}^I}. \tag{26}$$

As the average residence time in the $I$ compartment is dominated by recoveries we assume $T_{\text{res}}^I = 10$ days [54–56].

## Modeling vaccine efficacies

We assume the main effect of vaccinations on the individual to be twofold. A fraction $\eta$ that has received both vaccine doses will develop total immunity and not contribute to the spreading dynamics. The rest may, in principle, be infected with the virus but still have some protection against a severe course of the illness, resulting in a lower probability of dying or going to ICU. Both effects combined give the total protection against severe infections seen in vaccine studies, which we will denote with $\kappa$. For current COVID-19 vaccines, efficacies against severe disease $\kappa$ ranging from 70–99% [23, 31–33, 57–59] and infection blocking potentials $\eta$ of 60–90% [24, 41, 60, 61] are reported. The roughly uniform distribution of vaccine types in the European Union (see also Fig 8), consists to a larger part of mRNA-type vaccines for which comparatively high values $\kappa$ of 97–99% [33, 59] and $\eta$ of 80–90% are reported. We thus chose the rather conservative 90% for $\kappa$ and 75% for $\eta$ as our default values. The explicit $\kappa$ and $\eta$ do not explicitly appear in our equations, but as parameters $\eta_0$ and $\kappa_0$, which we derive from the reported numbers as follows.

Due to the lack of solid evidence on the effects of the first dose, we assume that the fraction of individuals developing total immunity already after the first dose is given by $\eta_0$. We further assume that of the $(1 - \eta_0)$ people that do not develop the immunity after the first dose, the same fraction $\eta_0$ acquires it after the second dose, i. e. the total vaccination path of the people that do not develop total immunity after both doses is given by $S_i \xrightarrow{1-\eta_0} S_i^1 \xrightarrow{1-\eta_0} S_i^2$. $\eta_0$ can thus be

related to $\eta$ by the formula

$$
\begin{aligned}
\eta \quad &= 1 - \frac{\text{not fully protected}}{\text{total vaccinated}} \\
&= 1 - (1 - \eta_0)^2 = \eta_0(2 - \eta_0).
\end{aligned}
\tag{27}
$$

For individuals vaccinated with both doses without total immunity, i. e., from $S_i^2$, we reduce the probabilities to die or go to ICU after infection to account for the reduced risk of severe symptoms due to the vaccine. Of the total number of people who get vaccinated the risk of going to ICU or dying is thus reduced by a factor

$$
1 - \kappa = (1 - \eta) \cdot (1 - \kappa_0),
\tag{28}
$$

from which we can deduce the value of $\kappa_0$.

Again, due to lack of solid data on the first doses we assume the risk of severe COVID-19 is reduced to a factor $\sqrt{(1 - \kappa)}$ when only a single dose has been received. From these assumptions we arrive at

$$
\delta_i^v = \left(\sqrt{1 - \kappa_0}\right)^v \delta_i,
\tag{29}
$$

$$
\alpha_i^v = \left(\sqrt{1 - \kappa_0}\right)^v \alpha_i,
\tag{30}
$$

$$
\gamma_i^v + \delta_i^v + \alpha_i^v = \bar{\gamma},
\tag{31}
$$

where $v = \{1, 2\}$ represents the dose of the vaccine for which an individual has successfully developed antibodies. Note that $v$ is used as a super-index on the left-hand side of the equation but as an exponent on the right-hand side. Eq 31 enforces vaccination not to alter the total average timescale of the disease course.

The transition rates from ICU to death, $\delta_i^{\text{ICU}}$, and from ICU to recovered, $\gamma_i^{\text{ICU}}$, are assumed to remain equal across doses. The reasons for this assumption are i) a lack of solid evidence for significant differences, and ii) once in ICU, it is reasonable to assume that the vaccine failed to work for this individual.

In addition to the effects of complete sterilizing immunity ($\eta$) and protection against severe disease ($\kappa$), we include a third effect of vaccines: Individuals that happen to have a breakthrough infection despite being vaccinated carry a lower viral load and are consequently less infectious than unvaccinated infected individuals. This has been shown already after the first dose [25, 60]. We include this effect by a factor $\sigma$ in the contagion term (cf. (1)).

### Individuals becoming infectious while developing antibodies

One special case that one has to consider is when individuals acquire the virus in the time frame between being vaccinated and developing an adequate antibody level. We assume that individuals share behavioral characteristics with the members of the corresponding susceptible compartment, so contagion follows the same dynamics. Let $X_i(s)$ be the fraction of susceptible individuals of a given age group vaccinated at time $s_0 < s$ and are not infected until time $s$. Assuming they can only leave the compartment by getting infected, the differential equation

governing their dynamics is:

$$\frac{dX_i}{ds} = -R_t X_i \sum_{j,v} C_{ji} \frac{\sigma^v I_j^v}{M_j} - \frac{X_i}{M_i}\Phi_i, \qquad \text{with } X_i(s_0) = 1. \qquad (32)$$

The solution of (32) is given by

$X_i(s) = \exp\left(-\int_{s_0}^{s} \sum_{j,v} R_{s'} C_{ji} \frac{\sigma^v I_j^v(s')}{M_j} ds'\right) \exp\left(-\frac{\Phi_i(s - s_0)}{M_i}\right)$. Following the same formalism

for every batch of vaccinated individuals produced at time $t - \tau$, the ones that remain suscepti-
ble by time $t$ are given by:

$$X_i(t) = \exp\left(-\int_{t-\tau}^{t} \sum_{j,v} R_{t'} C_{ji} \frac{\sigma^v I_j^v(t')}{M_j} dt'\right) \exp\left(-\frac{\Phi_i \tau}{M_i}\right). \qquad (33)$$

Therefore, we define the fraction of susceptible individuals acquiring the virus in the time-
frame of antibodies development as

$$p_i(t) = 1 - \exp\left(-\int_{t-\tau}^{t} \sum_{j,v} R_{t'} C_{ji} \frac{\sigma^v I_j^v(t')}{M_j} dt'\right) \exp\left(-\frac{\Phi_i \tau}{M_i}\right). \qquad (34)$$

This fraction is then subtracted in the transitions $V_i^v \rightarrow S_i^{v+1}$ from the vaccinated to the
immunized pools in the differential equations.

## Effect of test-trace-and-isolate

At low case numbers and moderate contact reduction, the spreading dynamics can be miti-
gated through test-trace-and-isolate (TTI) policies [18, 19]. In such a regime, individuals can
have slightly more contacts because the overall low amount of cases enables a diligent system
to trace offspring infections and stop the contagion chains. In other words, efficient TTI would
allow for having a larger gross reproduction number $R_t$ without rendering the system unstable.
The precise allowed increase in $R_t$ is determined by i) the rate at which symptomatic individu-
als are tested, ii) the probability of being randomly screened, and iii) the maximum capacity
and fraction of contacts that health authorities can manually trace. When the different compo-
nents of this meta-stable regime break down, we observe a self-accelerating growth in case
numbers.

In our age-stratified model, we do not explicitly include TTI, given all the uncertainties that
arise from the age-related modifying factors. However, we use our previous results to estimate
the gross reproduction number $R_t$ that would produce the same observed reproduction num-
ber in the different regimes of i) no test or contact tracing, ii) strict testing criteria, iii) self-
reporting, and iv) full TTI. Doing so, we build an empirical relation to evaluating the contex-
tual stringency of the different strategies herein compared (namely, long-term stabilization at
high or low case numbers).

In the phase diagram of Fig 9 we illustrate the conversion methodology. Two different $R_t$
might produce the same observed reproduction number $\hat{R}_t^{\text{obs}}$, depending on the regime in
which they operate. Fitting all curves to an exponential function, and assuming that the largest
eigenvalue of the system (for all possibilities of testing and tracing) can be represented as a

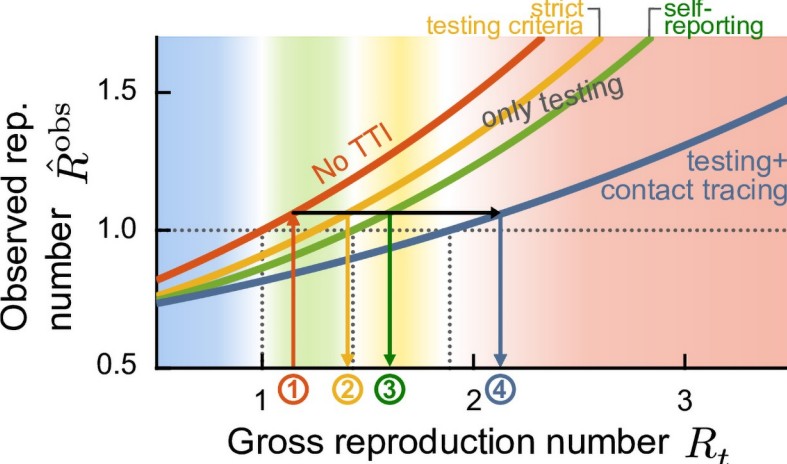

**Fig 9. Test-trace-and-isolate (TTI) policies allow for greater freedom (quantified by the gross reproduction number $R_t$) while observing the same reproduction number $\hat{R}_t^{\text{obs}}$.** Systematic efforts to slow down the spread of the disease, such as mass testing (random screening) and contact tracing, allow decreasing the observed reproduction number of the disease. For observing the same outcome in $\hat{R}_t^{\text{obs}}$, the gross reproduction number $R_t$ would increase, or, in other words, individuals would be allowed to increase their potentially contagious contacts. Therefore, we extrapolate the $R_t$ allowed in a full TTI setting at low case numbers and determine the equivalent $R_t$ trends required to reach the same $\hat{R}_t^{\text{obs}}$ in different regimes, starting from the raw value considering no TTI (red curve). Assuming that the relationship between $R_t$ and $\hat{R}_t^{\text{obs}}$ is exponential (Eq (35)), we can obtain the expected $R_t$ trends in the low-case numbers TTI regime. Starting from the raw $R_t$ curve (red, 1), we can obtain $R_t$ in all the other possible regimes: under strict testing criteria (yellow, 2), self-reporting (green, 3), or full TTI (blue, 4). Adapted from [18].

function of the gross reproduction number $R_t$, we obtain

$$\hat{R}_t^{\text{obs}} = a \exp\left(b R_t\right). \tag{35}$$

We then want to evaluate how to translate the values we get from our control problem (which has no testing nor tracing) to the equivalent in other regimes. Assuming that all strategies have the same $\hat{R}_t^{\text{obs}}$ (as schematized in Fig 9), we can relate their gross reproduction numbers in each regime through a simple equation:

$$R_t^i = \frac{1}{b_i}\left(\ln\left(\frac{a_0}{a_i}\right) + b_0 R_t\right), \tag{36}$$

which corresponds to a line, and where the subscript 0 represents the base scenario (with no testing or contact tracing) and the subscript $i$ represents the other strategies. The exponential fit to the curves shown in Fig 9 gives to the following line equations:

$$R_t^{\text{test(ineff)}} = 1.0211 R_t + 0.2229, \tag{37}$$

$$R_t^{\text{test(eff)}} = 1.0756 R_t + 0.3272, \tag{38}$$

$$R_t^{\text{TTI}} = 1.6842 R_t + 0.1805. \tag{39}$$

Assuming smooth transitions for these conversions in $R_t$, which are related to certain values of the new daily cases $N$ ($N_{\text{TTI}} < N_{\text{test(eff)}} < N_{\text{test(ineff)}} < N_{\text{no test}}$ respectively), we can define a

general conversion $R_t(N)$:

$$R_t(N) = \begin{cases} R_t^{\text{TTI}}, & \text{if } N < N_{\text{TTI}} \\ R_t^{\text{test(eff)}}\phi_1 + R_t^{\text{TTI}}(1-\phi_1), & \text{if } N_{\text{TTI}} \leq N < N_{\text{test(eff)}} \\ R_t^{\text{test(ineff)}}\phi_2 + R_t^{\text{test(eff)}}(1-\phi_2), & \text{if } N_{\text{test(eff)}} \leq N < N_{\text{test(ineff)}} \\ R_t\phi_3 + R_t^{\text{test(ineff)}}(1-\phi_3), & \text{if } N_{\text{test(ineff)}} \leq N < N_{\text{no test}} \\ R_t, & \text{else}, \end{cases} \tag{40}$$

where the $\phi$ parameters of each convex combination depend on $N$:

$$\begin{aligned} \phi_1 &= \frac{N - N_{\text{TTI}}}{N_{\text{test(eff)}} - N_{\text{TTI}}}, \\ \phi_2 &= \frac{N - N_{\text{test(eff)}}}{N_{\text{test(ineff)}} - N_{\text{test(eff)}}}, \qquad \text{and} \\ \phi_3 &= \frac{N - N_{\text{test(ineff)}}}{N_{\text{no test}} - N_{\text{test(ineff)}}}. \end{aligned} \tag{41}$$

Default reference values for the $N$-related set-points are $N_{\text{TTI}} = 20$, $N_{\text{test(eff)}} = 100$, and $N_{\text{test(ineff)}} = 500$ and $N_{\text{no test}} = 10000$ new daily cases per million. When we plot and refer to the gross reproduction number $R_t$, it is always the value obtained from Eq (40).

## Observed reproduction number

In real-world settings, the full extent of the disease spread can only be observed through testing and contact tracing. While the *true* number of daily infections $N$ is a sum of all new infections, the *observed* number of daily infections $\hat{N}^{\text{obs}}$ is the number of new infections discovered by testing, tracing, and surveillance of the quarantined individuals' contacts. Thus, the observed number of daily infections is given by

$$\hat{N}^{\text{obs}}(t) = \left[ \underbrace{\sum_{i,v} \rho E_i^v(t)}_{\text{end of latency}} + \underbrace{\sum_{i,v} \frac{S_i^v(t) + V_i^v(t)}{M_i} \Phi_i(t)}_{\text{ext. influx}} \right] \circledast \underbrace{\mathcal{K}(t)}_{\text{delay kernel}} \tag{42}$$

where $\circledast$ denotes a convolution and $\mathcal{K}$ an empirical probability mass function that models a variable reporting delay, inferred from German data. As the Robert-Koch-Institute (RKI), the official body responsible for epidemiological control in Germany [62], reports the date the test is performed, the delay until the appearance in the database can be inferred. The laboratories obtain 50% of the sample results on the next day, 30% the second day, 10% the third day, and further delays complete the remaining 10%, which for simplicity we will truncate at day four. Considering that an extra day is needed for reporting the laboratory results, the probability mass function for days 0 to 5 is given by $\mathcal{K} = [0, 0, 0.5, 0.3, 0.1, 0.1]$.

The spreading dynamics are usually characterized by the observed reproduction number $\hat{R}_t^{\text{obs}}$, an estimator of the effective reproduction number, calculated from the observed number of new cases $\hat{N}^{\text{obs}}(t)$. We use the definition underlying the estimates that are published by the RKI, which defines the reproduction number as the relative change of daily new cases

separated by 4 days (the assumed serial interval of COVID-19 [63])

$$\hat{R}_t^{\text{obs}} = \frac{\hat{N}^{\text{obs}}(t)}{\hat{N}^{\text{obs}}(t-4)}. \tag{43}$$

In contrast to the original definition of $\hat{R}_t^{\text{obs}}$ [62], we do not need to remove real-world noise effects by smoothing this ratio. It should be noted that calling $\hat{N}^{\text{obs}}$ the observed case numbers is somewhat misleading since we do not model the hidden figure explicitly. However, as this is expected only to change slowly, it is still sufficiently accurate to obtain the observed reproduction number from Eq (43).

## Keeping a steady number of daily infections with a PD control approach

With increasing immunity from the progressing vaccination program, keeping the spread of COVID-19 under control will require less and less effort by society. We can use this positive effect to lower the infections by upholding the same NPIs or gradually lifting restrictions to keep daily case numbers or ICU occupancy constant.

We model the optimal lifting of restrictions in the latter strategy using a Proportional Derivative (PD) control approach. The gross reproduction number $R_t$ is changed at every day of the simulation depending on either the daily case numbers $\hat{N}^{\text{obs}}$ or the total ICU occupancy $\sum_{i,v}\text{ICU}_i^v$ such that the system is always driven towards a given set point. The change in $R_t$ is negatively proportional to both the difference between the state and the setpoint as well as the change of that difference in time. The former dependence increases the number of infections if the case numbers drift down while the latter punishes rapid increases of the case numbers, keeping the system from overshooting the target value. We omit a dependence on the cumulative error, as is usually done in a PD controller, as that would enforce oscillations around the setpoint and because the PD has proven to be sufficient for our purposes.

Since both the case numbers and the ICU occupancy inherently only react to changes in $R_t$ after a few days of delay, we can further improve the stability of the control by "looking into the future". The full procedure for every day $t$ of the simulation then follows:

1. Run the system for a time span $T$ using the current $R_t$.

2. Quantify the relative error $\Delta(t + T)$ of the system state at the end by the difference between the observed case numbers or the total ICU occupancy and the chosen set point divided by said set point.

3. Calculate $R_t$ for the next day according to

$$R_{t+1\,\text{day}} = R_t - \left( k_p \cdot \Delta(t+T) + k_d \cdot \frac{d\Delta}{dt}(t+T) \right),$$

where $k_p$ and $k_d$ denote constant control parameters listed in Table 6.

**Table 6. The PD control parameters depending on the objective.**

| control problem | preview time span $T$ | proportional $k_p$ | derivative $k_d$ |
| --- | --- | --- | --- |
| $\hat{N}^{\text{obs}}$ (close to set point) | 14 days | 0.06 | 3.0 |
| $\hat{N}^{\text{obs}}$ (away from set point) | 14 days | 0.06 | 1.2 |
| $\sum_{i,v}\text{ICU}_i^v$ (close to set point) | 14 days | 0.2 | 15.0 |
| $\sum_{i,v}\text{ICU}_i^v$ (away from set point) | 14 days | 0.2 | 7.0 |

4. Revert the system from the state at $t + T$ to $t + 1$ day and start again at 1.

We use the same control system to uphold the setpoint as we use to drive the system towards that state from the initial conditions. In a staged-control-like manner, we make the system more reactive to high slopes near the setpoint, i. e. increase $k_d$ when within 10% of the target. In this way, the system can drive up quickly to the target while preventing overreactions to the gradual immunization changes while hovering at the fixed value.

Scenarios 2–4 in the main text consist of a chain of these control problems, changing from controlled case numbers to controlled ICU occupancy at one of the vaccination milestones (Fig 3).

## Parameter choices

For the age stratification of the population and the ICU rates, we used numbers published for Germany (Table 4). We suppose that the quantitative differences to other countries are not so large that the result would differ qualitatively. When comparing ICU rates across countries, one has to bear in mind that the definition of what constitutes an intensive care unit can differ between countries. We chose our ICU limit of 65 per million as a conservative limit so that in Germany, around three-quarters of the capacity would still be available for non-COVID patients. This limit was reached during the second wave in Germany. Other countries in the EU might have fewer remaining beds for non-COVID patients at this limit, as Germany has a comparatively high *per capita* number of ICU beds available.

ICU-related parameters are calculated from 14043 hospitalizations reported by German institutions until October 26, 2020 Table 5, converted to transition rates from Table 1. All other epidemiological parameters, their sources, values, ranges, and units are listed in detail in Table 2.

The vaccine efficacy, as discussed previously, is modeled as a multiplicative factor of the non-vaccinated reference parameter. The dose-dependent multiplicative factor is chosen to be 90% in the default scenario, which is in the range of the 70 to 95% efficacy measured in phase 3 studies [57] of approved vaccines and in accordance with the 92% efficacy of the Pfizer vaccine found in a population study in Israel [23]. In addition, we analyzed different scenarios of vaccine uptake (namely, the overall compliance of people to get vaccinated according to the vaccination plan) because of its relevance to policymakers and different scenarios of the protection the vaccine grants against infections $\eta$. The latter has great relevance for assessing risks when evaluating restriction lifting.

## Initial conditions

The initial conditions are chosen corresponding to the situation in Germany at the beginning of March 2021. We assume a seroprevalence of 10% because of post-infection immunity across all age groups, i.e., $R_i(0) = 0.1 \cdot M_i \, \forall i$. The vaccination at the beginning is according to the vaccination schedule introduced before, which leaves 5.1 million doses administered initially and an initial vaccination rate of 168 thousand doses per day. This compares to the 6.2 million total and the around 150 thousand daily administered doses at the time [26]. The initial number of daily new infections is at 200 per million, and the number of individuals treated in ICU is at 30 per million with an age distribution as observed during the first wave in Germany (taken from [47]). From these conditions and the total population sizes of the age groups (Table 4) we infer the initial size of each compartment.

## Numerical calculation of solutions

The system of delay differential equations governing our model were numerically solved using a Runge-Kutta 4th order algorithm, implemented in Rust (version 1.48.0). The source code is available on GitHub https://github.com/Priesemann-Group/covid19_vaccination.

## Supporting information

**S1 Fig. Sensitivity analysis centered at default parameters (solid black lines), for the fourth scenario from the main text.** We vary central parameters of the model individually, while keeping all others at their respective default value. For assessing the sensitivity to the TTI efficacy we scale all the capacity limits $N_{\text{TTI}}$, $N_{\text{test(eff)}}$, $N_{\text{test(ineff)}}$ and $N_{\text{no test}}$ (see Methods) by a common ratio.
(TIF)

**S2 Fig. Contact structure can have a significant impact on the population immunity threshold.** We assume that infections are kept stable at 250 daily infections until all age groups have been vaccinated. Then most restrictions are lifted, leading to a wave if vaccine uptake has not been high enough (see Fig 4A). We measure the severity of the wave (quantified by the duration of full ICUs) for varying uptake and vaccine efficacies for different contact structures (see Fig 7A–7C). **A-C**: The duration of the wave (measured by the duration of full ICUs) depends on the vaccine uptake and on the effectiveness of the vaccine measured by its efficacy at preventing infection (shades of purple) and severe illness (vaccine efficacy, full vs dashed vs dotted). **D-F**: If some NPIs are kept in place (such that the gross reproduction number goes up to $R_t = 2.5$), ICUs would be prevented from overflowing even in some cases of lower vaccine effectiveness. If precautionary measures are dropped in all age groups, including schools (A,D) the required uptake to prevent a further severe wave is increased by about 10% when compared to our default scenario of some continued measures to reduce the potential contagious contacts in school settings (B,E) or to completely homogeneous contacts (C,F). Not all combinations of vaccine effectiveness are possible as the vaccine efficacy against severe illness is by definition larger as the protection against any infection at all.
(TIF)

**S3 Fig. EU countries with different demographics have very similar dynamics—But the required vaccine uptake to guard against further severe waves is most sensitive to the initial seroprevalence.** Extended version of Fig 5, including more combinations of vaccine efficacies. **A–D**: If releasing all measures to pre-COVID contacts, keeping only some measures aiming to cup the reproduction number at 3.5. **E–H**: If releasing all measures to pre-COVID contacts, keeping only some measures aiming to cup the reproduction number at 3.5 and halving the contagiousness of contacts at school ages.
(TIF)

**S4 Fig. Even with the emergence of the highly contagious B.1.1.7 variant vaccinations are a promising mid-term strategy against COVID-19. Staying at low case numbers can greatly increase the individual freedom, especially in the long-term.** Schematic outlook into the effects of vaccination and the B.1.1.7 variant of SARS-CoV-2 on the societal freedom in the EU in 2021 compared to 2020 (see also the caption for Fig 1A). In 2020, seasonality effects and efficient test-trace-and-isolate (TTI) programs at low case numbers allowed for stable case numbers with only mild restrictions during summer, until about September. In 2021, vaccinations are expected to allow for greater freedom, but also a more contagious variant (B.1.1.7) is prevalent across the EU. Efficient TTI at low case numbers would thus help lifting major restrictions

earlier. The exact transition period between the wild type and B.1.1.7 (light purple shaded area) varies regionally.
(TIF)

**S5 Fig. Lowering the case numbers without the most stringent restrictions opens a middle ground between freedom and fatalities and prevents a new wave in the long term. A–D**: Variation of the fourth scenario from the main text (see Fig 3), where moderate restrictions are kept in place in the long term (letting the gross reproduction number go up to 2.5, compared to 3.5 in the default scenarios). **E–H**: Variation of the fifth scenario from the main text (see Fig 2) avoiding the strict initial restrictions. Keeping the gross reproduction number at a moderate level (1.5) until the everyone above 60 has been offered vaccination allows to decrease case numbers steadily. Over the summer a slight gradual increase in the contacts is allowed and all NPIs expect for test-trace-and-isolate (TTI) and enhanced hygiene are lifted when everyone received the vaccination offer (increasing the gross reproduction number to 3.5). **I**: The variation of the fourth scenario initially allows for the same increase in freedom as all the main scenarios, but needs more restrictions in the long term. The variation of the fifth scenario calls for stricter NPIs in the mid-term, but grants high freedom after summer. **J,K**: Both proposals lead to low number of infections and fatalities. **L**: Projected vaccination rates (see Fig 2).
(TIF)

**S6 Fig. Long-term control strategies (low vaccine uptake, 70% among the vaccinable population) from main text Figs 2 and 3.** Scenarios using default protection against infection $\eta =$ 0.75 and **low vaccine uptake of 70%** among the adult population.
(TIF)

**S7 Fig. Long-term control strategies (default vaccine uptake, 80% among the vaccinable population) from main text Figs 2 and 3.** Scenarios using default protection against infection $\eta = 0.75$ and **default vaccine uptake of 80%** among the adult population.
(TIF)

**S8 Fig. Long-term control strategies (high vaccine uptake, 90% among the vaccinable population) from main text Figs 2 and 3.** Scenarios using default protection against infection $\eta =$ 0.75 and **high vaccine uptake of 90%** among the adult population.
(TIF)

**S9 Fig. Mirror of Fig 2, using a homogeneous contact structure.**
(TIF)

**S10 Fig. Mirror of Fig 3, using a homogeneous contact structure.**
(TIF)

**S11 Fig. Mirror of S5 Fig, using a homogeneous contact structure.**
(TIF)

**S12 Fig. Mirror of Fig 2, using an empirical pre-COVID contact structure.**
(TIF)

**S13 Fig. Mirror of Fig 3, using an empirical pre-COVID contact structure.**
(TIF)

**S14 Fig. Mirror of S5 Fig, an empirical pre-COVID contact structure.**
(TIF)

**S1 Table. Parameters for the three main different vaccine uptake scenarios for Finland.**
Uptakes and averages are to be understood across the eligible (16+) population. For German data see Table 2 in the main text. Italian and Czech data are to be found in S2 and S3 Tables respectively.
(XLSX)

**S2 Table. Parameters for the three main different vaccine uptake scenarios for Italy.** The averages are to be understood across the eligible (16+) population. For German data see Table 2 in the main text. Finnish and Czech data are to be found in S1 and S3 Tables respectively.
(XLSX)

**S3 Table. Parameters for the three main different vaccine uptake scenarios for the Czech Republic. The averages are to be understood across the eligible (16+) population. For German data see Table 2 in the main text.** Finnish and Italian data are to be found in S3 and S2 Tables respectively.
(XLSX)

**S1 Supplementary Note. Eigenvalues of the homogeneous contact matrix.** Here we demonstrate a general case for the eigenvalues of a homogeneous contact matrix, for which every column accounts for the fraction age-groups represent respect to the total population.
(PDF)

## Acknowledgments

We thank the Priesemann group for exciting discussions and for their valuable input. We thank Christian Karagiannidis for fruitful discussions about the age-dependent hospitalization, ICU and fatality rates.

## Author Contributions

**Conceptualization:** Simon Bauer, Sebastian Contreras, Jonas Dehning, Alvaro Olivera-Nappa, Viola Priesemann.

**Data curation:** Simon Bauer, Matthias Linden.

**Formal analysis:** Simon Bauer, Sebastian Contreras, Jonas Dehning, Matthias Linden.

**Funding acquisition:** Viola Priesemann.

**Investigation:** Simon Bauer, Sebastian Contreras, Emil Iftekhar, Sebastian B. Mohr.

**Methodology:** Simon Bauer, Sebastian Contreras, Jonas Dehning, Matthias Linden, Viola Priesemann.

**Project administration:** Viola Priesemann.

**Software:** Simon Bauer.

**Supervision:** Viola Priesemann.

**Validation:** Simon Bauer, Sebastian Contreras, Jonas Dehning, Matthias Linden, Sebastian B. Mohr, Alvaro Olivera-Nappa, Viola Priesemann.

**Visualization:** Simon Bauer, Sebastian Contreras.

**Writing – original draft:** Simon Bauer, Sebastian Contreras, Jonas Dehning, Sebastian B. Mohr.

**Writing – review & editing:** Simon Bauer, Sebastian Contreras, Jonas Dehning, Emil Iftekhar, Sebastian B. Mohr, Alvaro Olivera-Nappa, Viola Priesemann.

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
