## [Decision Letter · Decision Letter 0]

20 May 2021

Dear Dr. Priesemann,

Thank you very much for submitting your manuscript "Relaxing restrictions at the pace of vaccination increases freedom and guards against further COVID-19 waves in Europe" for consideration at PLOS Computational Biology.

As with all papers reviewed by the journal, your manuscript was reviewed by members of the editorial board and by several independent reviewers. In light of the reviews (below this email), we would like to invite the resubmission of a significantly-revised version that takes into account the reviewers' comments.

This manuscript simulates different scenarios combining vaccination and NPI strategies. The trajectories resulting from this exercise point to relevant results that can provide practical guidance to policy makers. All reviewers attest the merit of this manuscript from this perspective. Two reviewers, however, raise important points regarding the unsatisfactory account of heterogeneities among European countries and the lack of important modeling structural components such as a contact matrix. Both issues represent important limitations of the work presented and require further attention from the authors, preferably pointing to potential instances in which their conclusions cannot hold. In addition to the above, the reviewers' remaining issues should also receive careful attention from the authors.

We cannot make any decision about publication until we have seen the revised manuscript and your response to the reviewers' comments. Your revised manuscript is also likely to be sent to reviewers for further evaluation.

Sincerely,

Claudio José Struchiner, M.D., Sc.D.

Associate Editor

PLOS Computational Biology

Virginia Pitzer

Deputy Editor-in-Chief

PLOS Computational Biology

This manuscript simulates different scenarios combining vaccination and NPI strategies. The trajectories resulting from this exercise point to relevant results that can provide practical guidance to policy makers. All reviewers attest the merit of this manuscript from this perspective. Two reviewers, however, raise important points regarding the unsatisfactory account of heterogeneities among European countries and the lack of important modeling structural components such as a contact matrix. Both issues represent important limitations of the work presented and require further attention from the authors, preferably pointing to potential instances in which their conclusions cannot hold. In addition to the above, the reviewers' remaining issues should also receive careful attention from the authors.

Reviewer's Responses to Questions

**Comments to the Authors:**

Reviewer #1: The authors have presented an age stratified compartmental model with a suitable structure to simulate COVID dynamics. This has been used to present a compelling argument for maintaining high levels of NPI measures until case numbers are reduced to a low level and vaccination programs are well advanced, which I believe to be well worth publishing.

My most major criticism is the suggestion that the modeling is "European". While many aspects of the model are certainly applicable most European countries (and even a majority of countries globally) there are also many important heterogeneities between European countries which really need to be considered for this to be presented as a European study, including: Social dynamics and population demographics-- something like Prim et als. mixing matrices should ideally be used; Vaccination speed and coverage-- the UK for instance is very advanced in its program, while many Eastern European countries are not; vaccine uptake-- again the UK for instance is already pushing 95% coverage in the over 50s, much higher than the central assumptions presented in this study which are based on German projections; vaccine types used-- the study here assumes a 75% efficacy against infection, which may be rather high for countries mainly using the Oxford/Az vaccine; TTI effects-- these form a key part of the results, but track and trace systems have been employed very differently across many countries and the results would really value some fitting to determine actual effects.

Model parameters for the study have been taken from a variety of sources, predominantly informed by German data, and while estimates seem reasonable ball parks in most cases i believe this would be a much stronger study if Germany was to be concentrated on as a case study with some evidence of model fitting to actual data preferably with parameter ranges to represent confidence intervals. To maintain a European angle, sensitivity could be presented with values indicated for different countries-- though I recognise this may be quite a lot of additional work. At the least it would be nice to see some support for TTI parameters as these seem intrinsic to even schematic results.

A few smaller points. In figure 2 j,k (and similar panels in other figures) totals are quoted for vaccination period given differing vaccine uptake. I found this quite a misleading comparison given that the period itself is dependent on uptake. Could this be presented instead as totals across 2021 (or some similar fixed period)?

I also noticed a few typos: Line 93 "for begin of March"; line 119 "update"; line 620/621 "because by getting".

Reviewer #2: The authors use a well-validated SEIRD-ICU model to explore the dynamics of balancing increased vaccination with increased mobility in a European setting. They find that, with the restriction that ICU limits are not exceeded, a relatively narrow range of scenarios is possible, since too-fast relaxation will quickly result in overwhelmed ICUs. The recommendation is to keep case counts as low as possible to facilitate contact tracing and reduce deaths.

Overall, I thought this was a carefully thought out and executed paper, with important findings that quantify just how narrow the range of policy options is: specifically (comparing Fig. 2a and 2e), the highest-infection and lowest-infection scenarios considered are virtually identical except for a 6-week period at the beginning as vaccination is ramping up.

Although the figures are complex, they are extremely well presented and so can still be easily followed. I tried running the scripts to regenerate them, but was not quite sure how to map the two outputs generated onto the figures in the manuscript. Additional documentation may be helpful on this point.

MINOR COMMENTS

p. 2, line 35: Perhaps add some qualification to the geographical scope of compliance/fatigue -- to my knowledge compliance remains high in some settings, such as Australia, Vietnam, and, recently, Bangladesh, which appears to have reversed its enormous spike in cases via a national lockdown.

p. 4, line 90: You mention 4% of the population already vaccinated in the simulations, but what about seroprevalence/cumulative incidence?

p. 5, Table 1: It's really interesting to see these data -- I hadn't realized that ICU-FR has such a strong age trend. One small point of confusion for me: I would have thought ICU-FR = (IFR) / (ICU probability), but this isn't the case, and indeed the IFR for >80 is higher than the ICU probability. While perhaps not the case in resource-limited settings, in a place like Germany I would've expected the number of ICU cases to exceed the number of deaths for all age groups.

p. 6, line 199: typo, "vaccine update" (uptake?)

p. 10, Figure 2: In panels j/k, I'm not sure I understand why "total infected" increases with vaccine uptake, or the meaning of the inset charts. Otherwise, this figure is beautifully clear despite its complexity!

p. 11, line 139: "complete relaxation of restrictions should wait until everyone has received a vaccination offer" -- by "everyone" do you mean adults? And by "ethical", do you mean in terms of equity? One could also ask, I suppose, whether it truly needs to be "everyone" or just above a herd immunity threshold.

p. 11, line 151: You may wish to consider the terms "non-susceptible" or "sterilizing immunity" rather than "sterile", which could have unintended negative connotations.

p. 11, line 153: There is increasing evidence of efficacy against variants, but true that much remains unknown. Perhaps soften this statement slightly.

Reviewer #3: In this manuscript the authors simulated different options for COVID-19 vaccination rollout in Germany, from March 2021, combined with the lifting of non-pharmaceutical intervention measures, subject to different constraints relating to the number of cases and healthcare capacity. The lifting of restrictions has been modelled by increasing the allowed number of contacts per person in the population. The authors aimed to identify the rate at which non-pharmaceutical interventions can be lifted, relative to the rate of vaccination. The central findings that the pace of vaccination determines the speed at which other restrictions can be lifted, and that vaccine uptake needs to be high to avoid further waves of infection, are supported by other studies. I found the manuscript interesting and enjoyed reading it, and all of the results and figures were thoughtfully put together. However I have identified some fundamental issues and questions with how the model is posed and described, that I will outline below.

Major comments

1. My main concern is with key terminology that affects the formulation of the transmission model. Throughout, the authors define R_t as the “relative number of contacts” or the “gross reproduction number”, which they have at one point described as being equivalent to the average number of infections generated from a single infected individual in a fully-susceptible population (which is the commonly understood definition of the reproduction number R), but then elsewhere refer to R_t as the number of contacts. This translates as though you are considering contacts and average new infections to be equivalent. Examining the differential equations, R_t seems to take the place of what would usually be formulated as the transmission term (conventionally a beta term to describe transmission) which accounts for both contacts and the risk of infection. Assuming that contacts and average new infections are equivalent is confusing and is not appropriate for SARS-CoV-2.

2. I do not agree that these results are broadly generalisable for Europe without considering the individual epidemic history and vaccination strategies of individual countries. The example and parameters in the manuscript are nearly all related to a single country, Germany, and it would not be appropriate to extrapolate these results across Europe (where epidemic trajectories and vaccination delivery do vary to date, although I appreciate there are some similarities in Western Europe), and this should be reflected in the title and text of the manuscript.

3. The authors note that they did not include a contact network structure in the transmission model, but instead assumed homogeneous contacts across all groups. Capturing a realistic age-based contact structure is critical for many infectious disease models, including for COVID-19, where the age-based distributions of those at highest risk of severe disease outcomes and those being vaccinated are non-uniform, and the interplay with realistic age-based contact patterns would therefore be important to include.

4. External infection term in the equations. I do not understand this part. Wouldn’t this typically be modelled as importations of infections, with increasing likelihood of importation as restrictions are gradually lifted? Looking at your differential equations in the Supplementary Material, susceptibles are being removed due to external contagion – but if those susceptibles are being infected when abroad, surely they should not be modelled as part of the local susceptible pool?

5. Throughout the manuscript you mention the strategy of maintaining low case numbers. It is important that you define what you mean by case numbers. Do you mean infections, symptomatic infections, individuals who have tested positive, or symptomatic individuals who have tested positive? These are quite different things – and will differ based on testing strategies.

6. The authors make the recommendation that “Throughout summer, only mild contact restrictions will remain necessary.” However, I am not convinced that this recommendation is founded by the research presented within this manuscript if numbers of permitted contacts are not explicitly related to non-pharmaceutical interventions. For example – how do specific measures such as school closures, versus mask-wearing, relate to the absolute number of contacts allowed?

7. Table 1. In O’Driscoll et al (Nature) the estimates of IFR by age are tabulated in 5-year age groupings. How did you translate these to the age groupings in column 2?

8. Immunity following natural infection. From my understanding, the authors assume no waning of immunity following natural infection. This is important over the timescales being considered, since the authors assume that 10% of the population has naturally-acquired immunity at the beginning of the simulation. A large proportion of that 10% would have been infected in the first wave and their immunity is likely waning (and is likely less protective against new circulating variants). The authors could consider performing a sensitivity analysis with lower levels of immunity assumed at March 2021.

9. In addition – did you explore sensitivity of your results to the values chosen for the external contagion term?

10. Figure 1a. You include a schematic representation of seasonality. Did you investigate seasonality of SARS-CoV-2 transmission in your study? I did not identify a seasonal forcing term in your equations. I don’t believe that we yet have a scientific consensus on the degree to which SARS-CoV-2 is seasonal, and the peaks observed in some countries in the southern hemisphere in summer (e.g. South Africa) would indicate that at least in the initial pandemic phase (before we approach some kind of endemic transmission), transmission could not be described as being strongly driven by seasonality.

11. The mechanism of vaccination needs to be explicitly explained. Lines 461-462. “There is some evidence that the vaccines partially prevent transmission of the disease. We therefore remove a fraction of vaccinated people completely from the disease dynamics by moving this fraction to the recovered compartments”. This assumes an “all-or-nothing” vaccine where a proportion of vaccinated individuals achieve complete sterile protection and are therefore removed, and that the remaining individuals have the same risk of infection as the unvaccinated population (but with lower risk of ICU and death). This is in contrast to a “leaky” vaccine where in the vaccinated population, there is a reduced risk of infection in line with observed efficacy against infection, and that individuals who experience a breakthrough infection then have a further reduced risk of severe disease/hospitalisation/ICU/death. In practice the reality is probably somewhere between the “all-or-nothing” and “leaky” extremes, but the dynamics will be different depending on which mechanism is used. I think what you have done is fine here but should be explained in the manuscript using conventional terminology.

Minor comments

1. Possibly a misunderstanding by me – but why is the denominator throughout the equations (S_i + R_i)?

2. Abstract. “despite most of the population still being susceptible” – this is a bit vague/imprecise as the level of susceptibility in the population depends on the protectiveness and duration of immunity following prior infection, the effectiveness of vaccines, and the number of people vaccinated.

3. Methods. You should specify that this is a deterministic, delay differential equation model formulated as a system of ordinary differential equations (I had to go to the supplement to understand what type of model you had used).

4. Line 71. Basic reproduction number – at what time point?

5. Line 72. “We further assume that the reproduction number can be decreased to about 3.5 by hygiene measures, face masks, and mild contact distancing.” Can you provide the logic for choosing this value?

6. Figure 1e. What do the shades of blue bars correspond to?

7. Line 90. March of 2021, I assume?

8. Line 92. “(assuming a case under-reporting factor of 2)” – do you have a source for this assumption?

9. S1.9. In the real world, the disease spread can only be observed through testing and tracing. This is not quite true. Disease would be observed through testing, yes, but also through hospitalisation and death records. The full extent of the spread of infection can be observed through testing and tracing.

10. Overall, I thought the figures were very good, however I found several of the figures difficult to interpret because there were too many panels within each figure (E.g. Figures 2 and 3, Figure S6), so the panels individually became quite small.

11. Figures S7 to S9 are missing the colour fill legend.

**Have the authors made all data and (if applicable) computational code underlying the findings in their manuscript fully available?**

Reviewer #1: Yes

Reviewer #2: Yes

Reviewer #3: Yes

PLOS authors have the option to publish the peer review history of their article (what does this mean?). If published, this will include your full peer review and any attached files.

Reviewer #1: No

Reviewer #2: **Yes: **Cliff Kerr

Reviewer #3: No
---

## [Decision Letter · Decision Letter 1]

14 Jul 2021

Dear Dr. Priesemann,

Thank you very much for submitting your manuscript "Relaxing restrictions at the pace of vaccination increases freedom and guards against further COVID-19 waves" for consideration at PLOS Computational Biology. As with all papers reviewed by the journal, your manuscript was reviewed by members of the editorial board and by several independent reviewers. The reviewers appreciated the attention to an important topic. Based on the reviews, we are likely to accept this manuscript for publication, providing that you modify the manuscript according to the review recommendations.

We suggest one more round of minor adjustments to the manuscript, as indicated by Reviewer 3, before its final acceptance.

Sincerely,

Claudio José Struchiner, M.D., Sc.D.

Associate Editor

PLOS Computational Biology

Virginia Pitzer

Deputy Editor-in-Chief

PLOS Computational Biology

[LINK]

We suggest one more round of minor adjustments to the manuscript, as indicated by two reviewers, before its final acceptance.

Reviewer's Responses to Questions

**Comments to the Authors:**

Reviewer #1: My thanks to the authors for addressing my previous concerns and comments. A thorough job seems to have been done with this and I have found the revised manuscript to be significantly improved, particularly in dealing with heterogeneities between countries and age groups. I would now be happy to recommend publication.

Reviewer #2: The authors have done an excellent job addressing the comments.

Reviewer #3: Thank you to the authors for submitting a substantially revised manuscript. I am satisfied with the responses to the comments and changes made to the manuscript, with the exception of my first major comment. I believe there remains an issue with nomenclature and commonly understood conventions in epidemiological modelling. I apologise if I did not make my comment clear initially.

The reproduction number R_t is described as the “gross reproduction number”, i.e. without the impact of immunity, to differentiate it from the effective reproduction number R_eff. This is fine. However, in the description, it is now stated that R_t quantifies the “average number of contacts that individuals have that potentially cause a COVID-19 infection in a fully susceptible population, i.e. the potentially contagious contacts”. This is incorrect. R_t is widely understood as the average number of new infections generated by a single infectious individual at time t. This is not the same as the average number of contacts that pose a risk of infection – the risk of infection to a susceptible person is driven by transmissibility of the pathogen as well as by the proportion of the population that is infectious (as reflected in your differential equations). Could the authors please correct this definition of the reproduction number?

**Have the authors made all data and (if applicable) computational code underlying the findings in their manuscript fully available?**

Reviewer #1: Yes

Reviewer #2: Yes

Reviewer #3: Yes

PLOS authors have the option to publish the peer review history of their article (what does this mean?). If published, this will include your full peer review and any attached files.

Reviewer #1: No

Reviewer #2: **Yes: **Cliff Kerr

Reviewer #3: No

Figure Files:

Data Requirements:

Reproducibility:

References:

---

## [Editor Report · Decision Letter 2]

19 Jul 2021

Dear Dr. Priesemann,

We are pleased to inform you that your manuscript 'Relaxing restrictions at the pace of vaccination increases freedom and guards against further COVID-19 waves' has been provisionally accepted for publication in PLOS Computational Biology.

Best regards,

Claudio José Struchiner, M.D., Sc.D.

Associate Editor

PLOS Computational Biology

Virginia Pitzer

Deputy Editor-in-Chief

PLOS Computational Biology

---

## [Editor Report · Acceptance letter]

10 Aug 2021

PCOMPBIOL-D-21-00605R2 

Relaxing restrictions at the pace of vaccination increases freedom and guards against further COVID-19 waves

Dear Dr Priesemann,

I am pleased to inform you that your manuscript has been formally accepted for publication in PLOS Computational Biology. Your manuscript is now with our production department and you will be notified of the publication date in due course.

With kind regards,

Livia Horvath
